# Proactive Cost Generation for Offline Safe Reinforcement Learning Without Unsafe Data

## Abstract

Learning constraint-satisfying policies from offline data without risky online interaction is essential for safety-critical decision making. Conventional approaches typically learn cost value functions from large numbers of unsafe samples in order to delineate the safety boundary and penalize potentially violating actions. In many high-stakes settings, however, risky trial-and-error is unacceptable, resulting in offline data that contains few, if any, unsafe samples. Under this data limitation, existing approaches tend to treat all samples as uniformly safe, neglecting the substantial presence of safe-but-infeasible states—states that are currently constraint-satisfying but inevitably violate safety constraints within a few future steps—thereby resulting in deployment failures. To overcome this challenge, we present PROCO, a proactive offline safe RL framework tailored to datasets largely devoid of violations. PROCO first learns a dynamics model from the offline data to capture environment information. It then constructs a conservative cost signal by grounding natural-language descriptions of unsafe states in large language models (LLMs), yielding risk assessments even when violations are unobserved. Finally, PROCO performs model-based rollouts under this cost to synthesize diverse and informative counterfactual unsafe samples, which in turn enable reliable feasibility identification and support feasibility-guided policy learning. Across diverse Safety-Gymnasium tasks, PROCO consistently reduces constraint violations and improves safety relative to conventional offline safe RL and behavior cloning baselines when training data contains only safe or minimally risky samples.

## 1 Introduction

Safe reinforcement learning (RL), which aims to derive policies that satisfy predefined safety constraints, is crucial for real-world applications such as autonomous driving (Zhang et al., 2021), robotic control (Brunke et al., 2022), and aligning large language models (LLM) with human values (Dai et al., 2024). A fundamental limitation of conventional online RL, is its reliance on trial-and-error exploration, which is inherently risky and can preclude its deployment in safety-critical settings (Levine et al., 2020). To mitigate these risks, offline safe RL has emerged as a promising paradigm, seeking to learn safe policies exclusively from pre-collected datasets without necessitating hazardous online interactions (Liu et al., 2024; Zheng et al., 2024).

In recent years, numerous methods have been proposed for offline safe policy learning. CPQ (Xu et al., 2022) and VOCE (Guan et al., 2023) learn conservative cost value functions to penalize unsafe actions, while COptiDICE (Lee et al., 2022) and FISOR (Zheng et al., 2024) leverage cost advantage functions as weighting to guide behavior cloning (BC) on offline data. From the policy optimization aspect, the effectiveness of both approaches critically relies on access to abundant unsafe samples, which enable accurate cost value estimation and subsequent policy optimization. However, in many safety-critical high-stakes scenarios, such as autonomous driving or robotics manipulation, the collection of large numbers of unsafe samples is impractical, as it is likely to cause detrimental effects on the agent or the environment Dulac-Arnold et al. (2019), leading to dataset with only few or no unsafe samples. In such cases, existing approaches may fail to derive policies that adhere to safety constraints, since without unsafe data, they generally consider all samples equally safe, neglecting the existence of **safe-yet-infeasible** samples. For instance, during robot data collection, to avoid harming the agent, external interventions are used to stop the process whenever the robot approaches a collision with an obstacle. As a result, states at the ends of these trajectories, though currently safe,

will inevitably lead to violations in the near future due to physical inertia if no external intervention is applied (Bansal et al., 2017). This limitation thus motivates our investigation into a new problem: **how can we learn a safe policy offline when unsafe samples are scarce or entirely absent?**

For the mentioned goal, we propose **Pro**active **Co**st Generation for Offline Safe Reinforcement Learning Without Unsafe Data (PROCO), a novel offline safe RL algorithm capable of identifying infeasible states and learning safe policies from datasets with few or no unsafe samples. PROCO first leverages an LLM to derive a conservative cost function from the natural language specification of the constraint, which is then validated and refined using the available safe data and, when present, a limited number of unsafe samples. Subsequently, it learns a dynamics model from the offline dataset. Using the cost function and dynamics model, PROCO simulates future state evolution to generate diverse and informative unsafe samples, enabling efficient detection of infeasible states. Moreover, the conservative cost function enables states in close proximity to unsafe ones to be labeled as unsafe as well, shortening the transition steps from infeasible to unsafe states and thus reducing the influence of model errors on infeasible state identification. Extensive experiments across diverse safety-gymnasium environments demonstrate that PROCO significantly outperforms existing offline safe RL and BC baselines in safe-only datasets, exceeding the best baseline by over 5 times in safety.

## 2 RELATED WORK

**Safe RL.** Safe RL seeks to maximize reward while ensuring safety, commonly modeled as a Constrained Markov Decision Process (CMDP) (Altman, 2021) and solved via constrained optimization (García & Fernández, 2015; Gu et al., 2024). A widely adopted approach is Lagrangian-based methods, which learn a cost value function with an adaptive multiplier to enforce safety (Stooke et al., 2020). For hard, state-wise constraints, Hamilton–Jacobi (HJ) reachability analysis (Bansal et al., 2017) has been applied to learn cost functions and enforce stricter safety (Yu et al., 2022; Ganai et al., 2023). However, both approaches require unsafe real-world interactions, which limits their practicality. To overcome this limitation, recent research has shifted toward offline safe RL, learning safe policies from pre-collected data to avoid unsafe exploration. Certain approaches integrate conservative value estimation into cost value function learning to counteract cost value underestimation (Xu et al., 2022; Guan et al., 2023), whereas others employ BC-based policy learning, including Decision Transformer (Liu et al., 2023), DICE (Lee et al., 2022), and IQL (Zheng et al., 2024; Koirala et al., 2025), to more effectively address extrapolation errors (Fujimoto et al., 2019) in safe RL.

**LLM assisted decision-making.** Leveraging the powerful information processing and reasoning capabilities of LLMs, they have recently been widely adopted in RL (Cao et al., 2024). One prominent line of work explores the use of LLMs for direct decision-making by generating actions or high-level plans conditioned on observations (Yao et al., 2023; Shinn et al., 2023; Prasad et al., 2023). Yet, these methods are generally applicable only to high-level, highly abstract decision-making tasks. A more broadly adopted line of work in RL involves reward function generation, in which LLMs are employed to directly construct reward functions that support skill discovery (Yu et al., 2023), policy learning (Song et al., 2023; Ma et al., 2024; Xie et al., 2024), exploration (Triantafyllidis et al., 2024), or teammate generation (Li et al., 2025a). However, the use of LLMs for cost function generalization in safe RL remains largely underexplored. More related work can be seen in Appendix C.

## 3 PRELIMINARIES

In this work, we focus on safe RL under hard constraints, which can be modeled as a hard constraint CMDP, defined as a tuple $\langle S, A, r, h, c, P, \gamma \rangle$. Here, $S$ and $A$ denote the state and action spaces, respectively; $r : S \times A \to [-R_{\max}, R_{\max}]$ represents the reward function, $h : S \to [h_{\min}, h_{\max}]$ is the constraint violation function, and $c : S \to \{0, 1\}$ is the cost function. $P : S \times A \times S \to [0, 1]$ specifies the transition dynamics, and $\gamma \in (0, 1)$ is the discount factor. Typically, $c(s) = \mathbb{I}(h(s) > 0)$, which means $h(s) > 0$ is unsafe while $h(s) \leq 0$ is safe. A policy $\pi : S \to \Delta(A)$ maps states to action distributions. Under policy $\pi$, the expected discounted reward return and cost return are defined as $R(\pi) = \mathbb{E}_{\tau \sim P_\pi} \left[ \sum_{t=0}^{\infty} \gamma^t r(s_t, a_t) \right]$ and $C(\pi) = \mathbb{E}_{\tau \sim P_\pi} \left[ \sum_{t=0}^{\infty} \gamma^t c(s_t) \right]$, where $\tau = (s_0, a_0, s_1, a_1, \dots) \sim P_\pi$ denotes a trajectory induced by $\pi$ and the environment dynamics $P$. Thus, the objective of solving a hard constraint CMDP is to find a policy that maximizes reward return while ensuring the cost return remains zero.

Figure 1: Structure of PROCO.

In the offline setting, we are given an offline dataset $\mathcal{D}$ generated by the behavior policy $\pi_\beta$. Now, the goal is to learning a safe policy purely from this dataset. To avoid extrapolation error in offline RL, the optimization objective is formulated as:

$$\max_\pi \ R(\pi), \ s.t. \ C(\pi) \leq 0 \ ; \ D(\pi||\pi_\beta) \leq \epsilon, \tag{1}$$

where $D(\pi||\pi_\beta)$ is a divergence term (e.g., KL divergence $D_{\mathrm{KL}}(\pi||\pi_\beta)$) used to prevent distribution shift. To investigate the problem of learning safe policies under scarce or absent unsafe samples, we focus on scenarios where $\mathcal{D}$ contains no unsafe samples. Optionally, we assume the availability of an extremely small dataset $\mathcal{D}_{\mathrm{unsafe}}$ consisting of no more than 100 unsafe transitions, satisfying $|\mathcal{D}_{\mathrm{unsafe}}| << |\mathcal{D}|$. Meanwhile, in most practical applications, a continuous constraint violation function $h$ is often unavailable. Accordingly, in this work we define $h$ as also a binary function: $h(s) = h_{\min} \leq 0$ if $c(s) = 0$, and $h(s) = h_{\max} > 0$ otherwise. To compensate for the lack of unsafe samples, we assume the availability of a natural language specification of the task's safety constraint, $L_{\mathrm{cost}}$, to provide safety-related information. Thus, the agent must exploit both the safe dataset $\mathcal{D}$ and the constraint description $L_{\mathrm{cost}}$ in order to acquire a safe policy.

## 4 METHOD

This section gives the detailed PROCO, a novel algorithm for learning safe policies offline when unsafe samples are scarce or entirely absent (Figure 1). Section 4.1 presents our dynamics model-based approach for feasibility identification, Section 4.2 introduces the motivation and methodology for employing LLMs to generate conservative cost functions, while Section 4.3 describes the overall pipeline of policy learning in PROCO. Implementation details can be found in Appendix D.

### 4.1 FEASIBILITY IDENTIFICATION WITH A DYNAMICS MODEL

Under the setting where only safe data are available and the task's cost function is assumed known, a key question arises: how can sample feasibility be assessed? To this end, we present a solution grounded in dynamics modeling and Hamilton–Jacobi (HJ) reachability analysis (Bansal et al., 2017).

We begin with a brief overview of the basic definitions of feasibility and HJ reachability analysis.

**Definition 4.1** (Feasible set and largest feasible set). The feasible set of a specific policy $\pi$ can be defined as

$$S_f^\pi := \{s \in S | h(s_t^\pi | s_0 = s) \leq 0, \forall t \in \mathbb{N}\}. \tag{2}$$

The largest feasible set $S_f^*$ is a subset of $S$ composed of states from which there exists at least one policy that keeps the system satisfying the constraint, i.e.,

$$S_f^* := \{s \in S | \exists \pi, h(s_t^\pi | s_0 = s) \leq 0, \forall t \in \mathbb{N}\}. \tag{3}$$

**Definition 4.2** (Optimal feasible value function). The optimal feasible state-value function $V_h^*$, and the optimal feasible action-value function $Q_h^*$ are defined as

$$V_h^*(s) := \min_\pi V_h^\pi(s) := \min_\pi \max_{t \in \mathbb{N}} h(s_t), s_0 = s, a_t \sim \pi(\cdot|s_t),$$

$$Q_h^*(s,a) := \min_\pi Q_h^\pi(s,a) := \min_\pi \max_{t \in \mathbb{N}} h(s_t), s_0 = s, a_0 = a, a_t \sim \pi(\cdot|s_t), \tag{4}$$

where $V_h^\pi$ represents the maximum constraint violations in the trajectory induced by policy $\pi$ starting from the state $s$. The (optimal) feasible value function possesses the following properties:

- $V_h^\pi(s) \le 0 \Rightarrow \forall s_t, h(s_t) \le 0$, indicating $\pi$ can satisfy the hard constraint starting from $s$. $V_h^*(s) \le 0 \Rightarrow \exists \pi, V_h^\pi(s) \le 0$, meaning there exists a policy that satisfies the hard constraint.

- The feasible set and largest feasible set can be rewritten as

$$S_f^\pi := \{s | V_h^\pi(s) \le 0\}, S_f^* := \{s | V_h^*(s) \le 0\}. \tag{5}$$

Based on Definition 4.1, once $S_f^*$ is obtained, the feasibility of states in $\mathcal{D}$ can be determined. Furthermore, Definition 4.2 indicates that $S_f^*$ can be obtained by computing the optimal feasible value function $V_h^*(s)$. Thus, the feasible Bellman operator $\mathcal{B}^*$ (Fisac et al., 2019) is proposed:

$$\mathcal{B}^* Q_h(s,a) := (1-\gamma)h(s) + \gamma \max\{h(s), V_h^*(s')\}, \ V_h^*(s') = \min_{a'} Q_h(s',a'). \tag{6}$$

However, computing $V_h^*(s)$ via $\mathcal{B}^*$ demands a substantial amount of unsafe samples beyond $\mathcal{D}$. To overcome this limitation, we propose leveraging a learned dynamics model to proactively generate future unsafe samples. First, we train an ensemble dynamics model $\hat{T}$ using the offline dataset $\mathcal{D}$:

$$\min_{\hat{T}} \mathbb{E}_{(s,a,s') \sim \mathcal{D}}[||\hat{T}(s,a) - s'||_2^2]. \tag{7}$$

Subsequently, leveraging $\hat{T}$ together with the dataset $\mathcal{D}$, we perform branched rollout to simulate future trajectories. To address the potential underestimation of $V_h^*$ due to model uncertainty, we introduce a conservative feasible Bellman operator, denoted as $\bar{\mathcal{B}}^*$:

$$\bar{\mathcal{B}}^* Q_h(s,a) := (1-\gamma)h(s) + \gamma \max\{h(s), \max_{s' \in \hat{T}(s,a)} \min_{a'} Q_h(s',a')\}, \tag{8}$$

For $\bar{\mathcal{B}}^*$, we establish the following desirable proposition:

**Proposition 4.3.** $\bar{\mathcal{B}}^*$ *is a $\gamma$ contraction mapping in the $\infty-$norm, and satisfies that $\bar{Q}_h^*(s,a) \ge Q_{h,T}^*(s,a)$ for all $(s,a)$ and for all $T \in \hat{T}$, where $\bar{Q}_h^*(s,a)$ is the convergence result of $\bar{\mathcal{B}}^*$ and $Q_{h,T}^*(s,a)$ is the convergence result of $\mathcal{B}^*$ under transition dynamics $T$.*

Finally, under certain assumptions, we can guarantee the effectiveness of feasibility identification when employing $\bar{\mathcal{B}}^*$ together with $\hat{T}$:

**Assumption 4.4.** There exists a horizon $H^* \in \mathbb{N}$ such that, for any infeasible state $s$, any sequence of actions $a_0, \ldots, a_{H^*-1}$ will lead to an unsafe state.

**Assumption 4.5.** The learned ensemble dynamics model $\hat{T}(s,a)$ is calibrated, that is, for all $(s,a) \in S \times A$, the ground truth dynamics model $T(s,a)$ satisfies that $T(s,a) \in \hat{T}(s,a)$.

**Theorem 4.6.** *If Assumption 4.4 and Assumption 4.5 hold, $\forall s \in \bar{S}_f^* = S - S_f^*, \forall a$, the convergence result $\bar{Q}_h^*$ learned by $\bar{\mathcal{B}}^*$ and $\hat{T}$ rollout data satisfies that, $\forall a, \bar{Q}_h^*(s,a) > 0$ for large enough $\gamma$.*

### 4.2 LLM ASSISTED CONSERVATIVE COST FUNCTION GENERATION

Theorem 4.6 enables feasibility identification for samples in $\mathcal{D}$, yet it critically depends on Assumption 4.5, which requires exact accuracy of the learned dynamics transitions, a condition rarely satisfied in practice. We therefore relax it as Assumption 4.7 and introduce another assumption:

**Assumption 4.7.** For a given policy $\pi$, there exists $\delta$, $\forall s, a, \max_t \mathbb{E}_{s \sim p_1^t(s)} D_{\text{KL}}(p_1(s'|s)||p_2(s'|s)) \le \delta$, where $p_1(s'|s) = T(s'|s, \pi(s))$ for $t > 0$ and $p_1(s'|s) = T(s'|s, a)$ for $t = 0$, and $p_2(s'|s) = T'(s'|s, \pi(s))$ for $t > 0$ and $p_2(s'|s) = T'(s'|s, a)$ for $t = 0$, $T$ is the ground truth dynamics model, and $T'$ refers to any one of the dynamics model in the ensemble dynamics model set $\hat{T}$.

**Assumption 4.8.** For a given constraint violation function $h$, there exists a constant $K$, such that for any two state distributions $\mu, v$, it satisfies that $|\mathbb{E}_\mu[h(s)] - \mathbb{E}_v[h(s)]| \le K D_{\text{TV}}(\mu||v)$, where $D_{\text{TV}}$ refers to the total variance distance.

Assumption 4.8 assumes a correlation between the expected constraint violation and the state distribution, such that similar state distributions correspond to similar expected constraint violations. This assumption is generally valid in practice, and based on it, we can proceed to analyze the effectiveness of feasibility identification:

**Theorem 4.9.** *If Assumption 4.4, Assumption 4.7 and Assumption 4.8 hold.* $\forall s \in \bar{S}_f^* = S - S_f^*$, $\forall a$, *given policy* $\pi$, *the convergence result* $\bar{Q}_h^*$ *learned by* $\bar{\mathcal{B}}^*$ *and branched rollout data of* $\pi$ *in* $\hat{T}$ *satisfies*

$$\bar{Q}_h^*(s,a) \geq (1 - \gamma^{t^*})h_{min} + \gamma^{t^*}\mathbb{E}_{p_1^{t^*}}[h(s)] - \gamma^{t^*}t^*K\delta, \ t^* = \arg\max_{t \in \{1,\ldots,H^*\}}[\mathbb{E}_{p_1^t}[h(s)]]. \quad (9)$$

**Remark.** Since $\gamma, h_{\min}, K$ are outer given of constant, Theorem 4.9 reveals that the feasible value funtion of the infeasible state is lower bounded by $t^*$, $h(s)$ and $\delta$. To ensure safe decision making, we hope $\bar{Q}_h^*(s,a)$ can be large enough. Since $\mathbb{E}_{p_1^t}[h(s)]$ is upper bounded by $h_{\max}$, a practical way is to lower $\delta$ or lower $t^*$. Here, $\delta$ denotes the model rollout error. Due to cumulative error, shorter rollout horizons lead to lower errors. The term $t^*$ represents the number of steps required to transition from an infeasible state $s$ to its associated unsafe state $s'$. Thus, the distance between $s$ and $s'$ determines $t^*$, with smaller distances yielding smaller values of $t^*$. Hence, the analysis indicates that **bringing an infeasible state closer to its corresponding unsafe state reduces the rollout error $\delta$ as well as the step horizon $t^*$, resulting in more efficient feasibility identification**.

Therefore, we propose leveraging an LLM to generate a more conservative cost function, which **marks states near actual unsafe states also as unsafe**, effectively bringing infeasible states closer to unsafe states. Specifically, we first provide three natural language descriptions, $L_{\text{task}}$, $L_{\text{cost}}$, and $L_{\text{inst}}$, to the LLM, enabling it to generate the task's cost function $\bar{c}$:

$$\bar{c} = \text{LLM}(L_{\text{task}}, L_{\text{cost}}, L_{\text{inst}}), \quad (10)$$

where $L_{\text{task}}$ is task-related information, such as the meaning of states; $L_{\text{cost}}$ is the provided language description of the safety constraint; and $L_{\text{inst}}$ provides explicit instructions to the LLM, directing it to generate a cost function that is more conservative than the constraint described in $L_{\text{cost}}$.

Nevertheless, the cost function produced directly by the LLM may not be reliable. To mitigate this, we propose a validation-and-feedback mechanism leveraging both the small unsafe dataset $\mathcal{D}_{\text{unsafe}}$ and the safe dataset $\mathcal{D}$. Concretely, we begin by validating $\bar{c}$ on $\mathcal{D}_{\text{unsafe}}$, requiring 100% accuracy to guarantee that all unsafe samples are correctly identified, thereby eliminating safety risks. After this criterion is satisfied, we evaluate the proportion of safe samples in $\mathcal{D}$ that are classified as unsafe, which quantifies the conservativeness of $\bar{c}$. If this proportion lies within the hyperparameter-controlled range $[p_{\min}, p_{\max}]$, we deem the conservativeness acceptable and adopt $\bar{c}$ as the final cost function. Otherwise, we construct a feedback description $L_{\text{feed}}$ from the evaluation outcomes and feed it back into the LLM to guide the regeneration of $\bar{c}$:

$$\bar{c} = \text{LLM}(L_{\text{task}}, L_{\text{cost}}, L_{\text{inst}}, L_{\text{feed}}). \quad (11)$$

If no satisfactory $\bar{c}$ is obtained after exhausting the maximum number of LLM queries, we adopt as the final cost function the candidate that attains the highest accuracy on $\mathcal{D}_{\text{unsafe}}$ and exhibits a conservativeness level on $\mathcal{D}$ closest to the interval $[p_{\min}, p_{\max}]$.

## 4.3 OVERALL ALGORITHM

With the learned $\hat{T}$ and the obtained $\bar{c}$, we can now learn a feasible policy based on the safe-only dataset $\mathcal{D}$. First, we initialize a model rollout dataset $\mathcal{D}_{\hat{T}}$, and use the learning policy $\pi$ to perform branched rollouts of length $H$ within $\hat{T}$, obtaining a set of branch trajectories $\{\tau_i\}_{i=1}^N$. Each trajectory is of the form $\tau_i = (s_0^i, a_0^i, \bar{c}(s_0^i), \hat{r}(s_0^i, a_0^i), \ldots, s_{H-1}^i, a_{H-1}^i, \bar{c}(s_{H-1}^i), \hat{r}(s_{H-1}^i, a_{H-1}^i))$, where $s_0^i \in \mathcal{D}$, $s_t^i \sim \hat{T}(\cdot|s_{t-1}^i, a_{t-1}^i)$ for $t > 0$, and $\hat{r}$ denotes the reward model jointly learned with $\hat{T}$. Next, for any trajectory $\tau_i$, if it contains a safety violation, i.e., $\sum_{t=0}^{H-1} \bar{c}(s_t^i) > 0$, we add it to $\mathcal{D}_{\hat{T}}$. Otherwise, to reduce the impact of model errors on the stability of value function learning, trajectories without safety violations are discarded. To enhance data diversity during model rollouts and thereby improve the efficiency of feasibility identification, we inject Gaussian noise with standard deviation $\sigma_{\text{exp}}$ (a hyperparameter) to the policy actions during rollout. Meanwhile, the existing samples in $\mathcal{D}$ are further re-labeled with costs using $\bar{c}$. Finally, we derive the constraint violation labels according to $h(s) = h_{\min} \leq 0$ if $\bar{c}(s) = 0$, and $h(s) = h_{\max} > 0$ otherwise.

With the constraint violation labels in place, we approximate the minimization operator in the (conservative) feasible Bellman update, i.e., $V_h^*(s') = \min_{a'} Q_h(s', a')$, by employing reverse

expectile regression to update the constraint violation value network:

$$\mathcal{L}_{V_h} = \mathbb{E}_{(s,a)\sim\mathcal{D}\cup\mathcal{D}_{\hat{T}}}[\mathcal{L}_{\text{rev}}^{\tau}(Q_h(s,a) - V_h(s))],$$

$$\mathcal{L}_{Q_h} = \mathbb{E}_{(s,a,s')\sim\mathcal{D}}[((1-\gamma)h(s) + \gamma\max(h(s), V_h(s')) - Q_h(s,a))^2]$$

$$+ \mathbb{E}_{(s,a)\sim\mathcal{D}_{\hat{T}}}[((1-\gamma)h(s) + \gamma\max(h(s), \max_{s'\in\hat{T}(s,a)} V_h(s')) - Q_h(s,a))^2], \quad (12)$$

where $\mathcal{L}_{\text{rev}}^{\tau}(u) = |\tau - \mathbb{I}(u > 0)|u^2$, $\tau$ is a hyperparameter. For the reward value function, we update it using expectile regression in a manner analogous to IQL (Kostrikov et al., 2022):

$$\mathcal{L}_{V_r} = \mathbb{E}_{(s,a)\sim\mathcal{D}}[\mathcal{L}^{\tau}(Q_r(s,a) - V_r(s))],$$

$$\mathcal{L}_{Q_r} = \mathbb{E}_{(s,a,s',r)\sim\mathcal{D}}[(r + \gamma V_r(s') - Q_r(s,a))^2], \quad (13)$$

where $\mathcal{L}^{\tau}(u) = |\tau - \mathbb{I}(u < 0)|u^2$. Then, we employ a policy extraction approach in which the optimization objective is decomposed into feasible and infeasible components separately, a formulation that has proven to be effective in previous work (Zheng et al., 2024):

**Feasible:** $\max_{\pi} \mathbb{E}_{a\sim\pi}\left[A_r^*(s,a) \cdot \mathbb{I}_{V_h^*(s)\leq 0}\right]$     **Infeasible:** $\max_{\pi} \mathbb{E}_{a\sim\pi}\left[-A_h^*(s,a) \cdot \mathbb{I}_{V_h^*(s)>0}\right]$

$$\text{s.t.} \int_{\{a|Q_h^*(s,a)\leq 0\}} \pi(a|s)da = 1, \forall s \in \mathcal{S}_f^* \qquad \text{s.t.} \int_a \pi(\cdot|s)da = 1$$

$$D_{\text{KL}}(\pi||\pi_\beta) \leq \epsilon \qquad\qquad\qquad D_{\text{KL}}(\pi||\pi_\beta) \leq \epsilon$$

$$(14)$$

where $A^*(s,a) = Q^*(s,a) - V^*(s)$ are advantage functions. Leveraging Lagrangian multipliers and the KKT conditions, we can derive a closed-form solution to the above optimization objective:

$$\pi^*(a|s) = w(s,a)\pi_\beta(a|s), \, w(s,a) = \begin{cases} \frac{1}{Z_1(s)}\exp(\alpha_1 A_r^*(s,a)) \cdot \mathbb{I}_{Q_h^*(s,a)\leq 0} & V_h^*(s) \leq 0 \\ \frac{1}{Z_2(s)}\exp(-\alpha_2 A_h^*(s,a)) & V_h^*(s) > 0 \end{cases}, \quad (15)$$

where $Z_1(s)$ and $Z_2(s)$ are normalization terms that can be omitted, $\alpha_1$ and $\alpha_2$ are hyperparameters. Finally, policy extraction can be formalized as $\mathbb{E}_{(s,a)\sim\mathcal{D}}[w(s,a)\log\pi_\theta(s,a)]$. To enhance distribution modeling capacity, the policy $\pi_\theta$ can be parameterized with a diffusion model $\epsilon_\theta$, and optimized via the following objective:

$$\mathcal{L}_\theta = \mathbb{E}_{(s,a)\sim\mathcal{D},\epsilon,t}\left[\frac{\beta_t^2}{2\sigma_t^2\alpha_t(1-\bar{\alpha}_t)}w(s,a)||\epsilon - \epsilon_\theta(\sqrt{\bar{\alpha}_t}a + \sqrt{1-\bar{\alpha}_t}\epsilon, s, t)||^2\right], \quad (16)$$

where $\beta_t$ is human designed noised schedule, $\alpha_t = 1 - \beta_t, \bar{\alpha}_t = \prod_{s=1}^t \alpha_t, \sigma_t^2 = \beta_t$.

## 5 EXPERIMENTS

In this section, we present our experimental analysis conducted on 17 Safety-Gymnasium (Ji et al., 2023) tasks from the modified OSRL (Liu et al., 2024) dataset to answer the following questions: (1) Can PROCO outperform other baselines across various tasks, and how each design of PROCO contribute to its performance (Section 5.2)? (2) Do infeasible states in safe-only datasets compromise the safety of policy learning, and can PROCO effectively address this challenge (Section 5.3)? (3) What is the impact of different hyperparameter choices on PROCO 's performance (Section 5.4)?

### 5.1 BASELINES AND TASKS

To evaluate PROCO, we compare it with representative offline safe RL (also can be seen as offline RL) baselines. We first compare our method against CPQ (Xu et al., 2022) (CQL (Kumar et al., 2020)), aiming to examine how well conservative value estimation techniques perform in the context of learning safe policies offline with scarce or absent unsafe samples. Subsequently, to strengthen the ability of extrapolation error mitigation, we benchmark against BCQ Lagrange (BCQ) (Fujimoto et al., 2019), a batch-constrained baseline. Finally, we further compare against four BC-based baselines. Among them, BC and CDT (Liu et al., 2023) (DT (Chen et al., 2021)) update policies using the standard supervised BC loss, differing only in whether Return-to-Go and Cost-to-Go tokens

Table 1: Overall normalized rewards and costs. Each value is averaged over 20 evaluation episodes, and 3 random seeds. **Green**: The best-performing agent. **Blue**: The second best-performing agent.

| Task | CPQ | | BCQ Lagrange | | BC | | CDT | | COptiDICE | | FISOR | | PROCO | |
|---|---|---|---|---|---|---|---|---|---|---|---|---|---|---|
| | r↑ | c↓ | r↑ | c↓ | r↑ | c↓ | r↑ | c↓ | r↑ | c↓ | r↑ | c↓ | r↑ | c↓ |
| PointButton1 | 0.77 | 13.37 | 0.25 | 4.64 | 0.13 | 5.20 | 0.47 | 13.27 | 0.06 | 2.80 | 0.49 | 11.50 | 0.01 | 1.15 |
| PointButton2 | 0.62 | 14.53 | 0.41 | 8.65 | 0.28 | 6.68 | 0.59 | 15.27 | 0.14 | 4.86 | 0.46 | 14.69 | 0.08 | 2.76 |
| PointGoal1 | 0.80 | 5.39 | 0.71 | 3.89 | 0.58 | 3.30 | 0.75 | 5.01 | 0.50 | 5.34 | 0.73 | 5.70 | 0.35 | 0.96 |
| PointGoal2 | 0.80 | 18.69 | 0.66 | 11.57 | 0.53 | 8.07 | 0.78 | 13.90 | 0.38 | 5.15 | 0.59 | 17.10 | 0.08 | 0.38 |
| PointPush1 | 0.19 | 6.35 | 0.31 | 2.76 | 0.24 | 4.65 | 0.30 | 4.65 | 0.09 | 3.58 | 0.34 | 3.70 | 0.17 | 0.86 |
| PointPush2 | 0.22 | 10.21 | 0.24 | 4.03 | 0.22 | 3.70 | 0.28 | 4.51 | 0.03 | 3.00 | 0.27 | 6.61 | 0.12 | 0.30 |
| CarButton1 | 0.35 | 42.38 | -0.02 | 5.56 | -0.09 | 3.32 | 0.25 | 15.53 | -0.04 | 3.69 | 0.47 | 28.01 | -0.03 | 0.60 |
| CarButton2 | 0.54 | 34.06 | 0.00 | 5.15 | -0.09 | 4.64 | 0.35 | 19.98 | -0.08 | 2.93 | 0.57 | 25.96 | 0.00 | 0.99 |
| CarGoal1 | 0.80 | 5.74 | 0.47 | 2.86 | 0.36 | 2.15 | 0.71 | 5.04 | 0.38 | 2.14 | 0.75 | 4.10 | 0.13 | 0.02 |
| CarGoal2 | 0.82 | 20.41 | 0.25 | 3.67 | 0.22 | 2.96 | 0.67 | 11.43 | 0.30 | 3.72 | 0.80 | 14.56 | 0.03 | 0.52 |
| CarPush1 | -0.01 | 5.08 | 0.20 | 1.59 | 0.19 | 1.10 | 0.32 | 2.86 | 0.20 | 2.21 | 0.42 | 1.91 | 0.16 | 0.93 |
| CarPush2 | 0.29 | 16.36 | 0.15 | 6.94 | 0.08 | 4.14 | 0.23 | 9.88 | 0.07 | 3.00 | 0.39 | 8.04 | 0.01 | 0.04 |
| SwimmerVelocityV1 | 0.06 | 11.65 | 0.56 | 24.28 | 0.46 | 5.70 | 0.70 | 4.59 | 0.48 | 3.43 | -0.05 | 4.51 | 0.02 | 0.12 |
| HopperVelocityV1 | 0.06 | 5.09 | 0.36 | 7.66 | 0.35 | 1.33 | 0.63 | 3.44 | 0.18 | 4.59 | 0.07 | 3.23 | 0.11 | 0.07 |
| HalfCheetahVelocityV1 | 1.35 | 88.45 | 1.01 | 25.83 | 0.95 | 4.27 | 0.95 | 1.42 | 0.59 | 0.00 | 1.03 | 16.13 | 0.48 | 0.00 |
| Walker2dVelocityV1 | 0.03 | 1.08 | 0.79 | 0.24 | 0.69 | 2.67 | 0.79 | 0.55 | 0.10 | 1.68 | 0.15 | 5.71 | 0.09 | 0.58 |
| AntVelocityV1 | -1.01 | 0.00 | 0.92 | 27.89 | 0.98 | 11.54 | 0.99 | 1.85 | 0.98 | 5.16 | 1.01 | 10.47 | 0.52 | 0.00 |
| Average | 0.39 | 17.58 | 0.43 | 8.66 | 0.36 | 4.44 | 0.57 | 7.84 | 0.26 | 3.37 | 0.50 | 10.70 | 0.14 | 0.60 |

are included as inputs. In contrast, COptiDICE (Lee et al., 2022) and FISOR (Zheng et al., 2024) represent state-of-the-art (SOTA) offline safe RL algorithms built on advantage-weighted BC. During comparison, all baselines under soft-constraint modeling adopt 0 as the cost limit. For performance evaluation, reward returns are normalized using the minimum and maximum trajectory returns in the offline dataset, while costs are normalized by a factor of 10. Lower costs indicate better performance, and for cases with equal cost, higher rewards signify superior performance.

The tasks selected from Safety-Gymnasium used in our experiments consist of 12 navigation tasks and 5 velocity tasks. Specifically, the navigation tasks involve two types of robots (Point and Car) across three distinct scenarios: Button, Goal, and Push, with two tasks per scenario. On the other hand, the velocity tasks cover five different robots—Ant, HalfCheetah, Hopper, Swimmer, and Walker2d. To thoroughly demonstrate the effectiveness of PROCO, we select the most challenging OSRL dataset, where each task contains trajectories with varying degrees of constraint violations. By removing all unsafe data, we construct safe-only datasets that still cover diverse levels of feasibility. In addition, for each task, we retain 100 unsafe samples, which are reserved for validating the cost functions generated by the LLM. Additional details can be found in Appendix E.

## 5.2 COMPETITIVE RESULTS AND ABLATIONS

In this section, we first present the overall performance of PROCO and all baselines across different Safety-Gymnasium tasks, and the results are summarized in Table 1. First, CPQ exhibits the weakest safety performance, suggesting that conservative estimation methods provide insufficient constraints on the policy distribution relative to the offline dataset. As a result, the policy tends to deviate from the safety boundaries defined by the dataset, leading to severe safety violations. BCQ Lagrange, by leveraging a CVAE to impose stronger batch constraints, achieves clear safety improvements over CPQ. However, the number of violations remains substantial. In contrast, BC-based methods generally yield stronger average performance. CDT and FISOR, due to their use of generative models with strong distribution modeling capabilities, tend to overfit the reward maximization objective. While this yields clear gains in reward performance, their safety performance remains unsatisfactory.

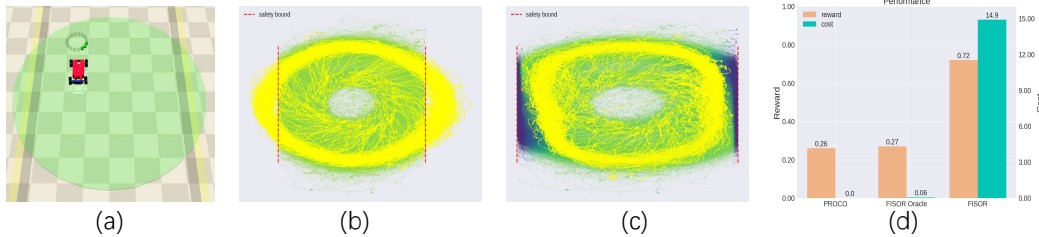

(a)          (b)          (c)          (d)

Figure 3: (a) Visualization of the Circle task. (b) and (c) Visualization of FISOR and PROCO performance, where the red dashed line denotes the safety bound, while the yellow dots indicate the state visitation distribution of the policy. The remaining points represent the value estimations of the constraint violation value function for samples in $\mathcal{D}$, where darker colors correspond to higher estimated violation values. (d) Final performance comparison of FISOR and PROCO.

Conversely, BC and COptiDICE, which employ supervised BC-style learning, more effectively constrain the policy within the offline distribution and thus achieve the best safety results among existing baselines. Nonetheless, they cannot differentiate between feasible and infeasible regions within the dataset, often leading to frequent visits to infeasible states and ultimately limiting their overall safety performance. By comparison, only PROCO, through the integration of a dynamics model and a conservative cost function, achieves effective feasibility identification, resulting in a substantial enhancement of safety performance—surpassing the strongest baseline by over five times.

Next, we conduct ablation studies to evaluate the contributions of key components in PROCO. The following variants are considered: (1) **W/o Model** omits the use of a dynamics model for generating unsafe samples, instead solely applying the conservative cost function to relabel the offline dataset. (2) **Full Model** utilizes all model rollout data to update all the value functions and the policy. (3) **W/o Relabel** applies the conservative cost function only to label the model rollout data and does not relabel the offline dataset. (4) **Det. Rollout** executes model rollouts deterministically with the policy, omitting the injection of extra noise. (5) **W/o Consv.** does not require the LLM to generate a cost function that is more conservative than the given safety constraint description. (6) **W/o Refl.** does not use the additional unsafe samples $\mathcal{D}_{unsafe}$ or the safe

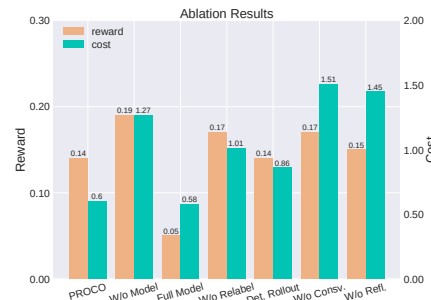

Figure 2: Ablation study results.

offline dataset to validate and provide feedback on the correctness and conservativeness of the LLM-generated cost function. As shown in Figure 2, when the dynamics model is not used, safety performance drops significantly, confirming the effectiveness of model-based feasibility identification. On the other hand, employing all model rollout data for updates to all value functions and the policy preserves high safety performance but significantly reduces reward performance, suggesting that errors accumulated during model rollouts can destabilize value learning and negatively impact reward estimation. Meanwhile, the decline in safety performance for W/o Relabel and Det. Rollout confirms the effectiveness of relabeling offline data with the conservative cost function and adding extra noise during model rollouts to enhance data diversity. Finally, the poor safety performance of W/o Consv. and W/o Refl. highlights the importance of generating a more conservative cost function for effective feasibility identification, as well as the necessity of validating and refining the LLM-generated cost function using existing data. More detailed results are provided in Appendix F.

## 5.3 CASE STUDY

Here, we aim to verify whether infeasible states in a safe-only dataset can adversely affect safe policy learning. To this end, we conduct a simple visualization experiment on the Ant Circle task. As illustrated in Figure 3(a), the Circle task requires the agent to move along the circumference of a circle as closely as possible, while crossing the left or right boundaries is considered unsafe.

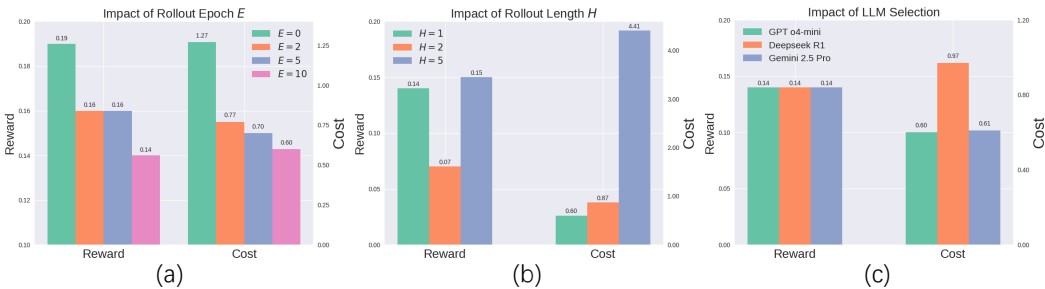

Figure 4: (a) Sensitivity analysis on rollout epoch $E$. (b) Sensitivity analysis on rollout length $H$. (c) Sensitivity analysis on LLM selection.

In this task, we compare the BC-based SOTA offline safe RL algorithm FISOR against PROCO. As illustrated in Figure 3(b), when no unsafe samples are available, FISOR's constraint violation value function fails to distinguish the feasibility of samples in the offline dataset. Consequently, the learned policy essentially prioritizes reward maximization. Even with BC-based policy extraction, the policy inevitably deviates from the offline distribution, resulting in a large number of unsafe samples being generated out of distribution. In contrast, as shown in Figure 3(c), PROCO leverages the conservative cost function together with the dynamics model to successfully identify samples near the safety boundary as infeasible. This enables the policy to adjust its behavior in time, preventing it from drifting beyond the safe region of the offline dataset and thereby avoiding constraint violations. Finally, as shown in Figure 3(d), PROCO successfully achieves zero safety violations, attaining performance comparable to FISOR Oracle (which is trained with access to a large amount of additional unsafe samples), whereas FISOR produces a substantial number of unsafe behaviors.

## 5.4 SENSITIVITY ANALYSIS

Finally, we investigate the impact of different hyperparameter choices on PROCO 's performance. During model rollouts, three hyperparameters play a crucial role: the rollout batch size $b$, the number of rollout epochs $E$, and the rollout length $H$. Specifically, $b$ denotes the number of samples drawn, $H$ is the number of steps rolled out per sample, and $E$ indicates how many times previous process is repeated per rollout. Since both $b$ and $E$ determine the total amount of rollout data, we focus our analysis on $E$ and $H$. First, as shown in Figure 4(a), increasing $E$ leads to a noticeable decline in reward and cost. This indicates that a larger amount of rollout data enables more accurate cost value learning, further confirming the effectiveness of model-based feasibility identification. Second, Figure 4(b) shows that as the $H$ increases, the policy's safety performance deteriorates, with a particularly sharp drop observed at $H = 5$. This suggests that increased model compounding error can introduce instability in cost value learning, leading to failures in feasibility identification. It also indirectly underscores the necessity of using a conservative cost function to reduce the distance from infeasible states to unsafe states. In the end, Figure 4(c) presents the effect of different LLMs on performance. GPT o4-mini and Gemini 2.5 Pro demonstrate comparable overall results, whereas Deepseek R1 shows lower safety performance. This suggests that the choice of LLM can impact PROCO 's effectiveness, with more capable LLMs generally yielding better outcomes.

## 6 FINAL REMARKS

In this work, we propose PROCO, a novel algorithm for learning safe policies offline when unsafe samples are scarce or entirely absent. PROCO first leverages the offline dataset to learn a dynamics model, while simultaneously employing an LLM to generate a conservative cost function tailored to the task. Based on the learned dynamics model and cost function, it then performs branched rollouts from the offline data samples to simulate their potential future evolutions. The generated rollout data are further incorporated into HJ reachability analysis for feasibility identification, which in turn guides policy extraction to ensure both safety and effectiveness. Extensive experiments across diverse tasks demonstrate the superior safety performance of PROCO. In the future, leveraging more powerful Vision-Language Models (VLMs) to extend PROCO to visual tasks and embodied intelligence in a more cost-efficient manner represents a highly promising research direction.

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

## A  LLM USAGE

In our paper, LLMs constitute a crucial component of the proposed methodology. Furthermore, during the writing process, LLMs are employed to calibrate and polish language expressions.

## B  MATHEMATICAL PROOFS

### B.1  PROOF OF PROPOSITION 4.3

*Proof.* First, we aim to proof that

$$|\max(a_1, b_1) - \max(a_2, b_2)| \le \max(|a_1 - a_2|, |b_1 - b_2|). \tag{17}$$

To start, if $a_1 \ge b_1$, then

$$\max(a_1, b_1) - \max(a_2, b_2) = a_1 - \max(a_2, b_2) \le a_1 - a_2. \tag{18}$$

Otherwise, if $b_1 \ge a_1$, then

$$\max(a_1, b_1) - \max(a_2, b_2) = b_1 - \max(a_2, b_2) \le b_1 - b_2. \tag{19}$$

Therefore, we have

$$\max(a_1, b_1) - \max(a_2, b_2) \le \max(a_1 - a_2, b_1 - b_2). \tag{20}$$

Similarly, by switching the subscript 1 and 2, we have

$$\max(a_2, b_2) - \max(a_1, b_1) \le \max(a_2 - a_1, b_2 - b_1), \tag{21}$$

which is equivalent to

$$-(\max(a_1, b_1) - \max(a_2, b_2)) \le \max(-(a_1 - a_2), -(b_1 - b_2)). \tag{22}$$

By observing that $\max(x, y) \le \max(|x|, |y|)$, we have

$$-\max(|a_1 - a_2|, |b_1 - b_2|) \le \max(a_1, b_1) - \max(a_2, b_2) \le \max(|a_1 - a_2|, |b_1 - b_2|), \tag{23}$$

which means

$$|\max(a_1, b_1) - \max(a_2, b_2)| \le \max(|a_1 - a_2|, |b_1 - b_2|). \tag{24}$$

Therefore, we have

$$\|\bar{\mathcal{B}}^* Q - \bar{\mathcal{B}}^* Q'\|_\infty = \sup_{s,a} |\bar{\mathcal{B}}^* Q(s, a) - \bar{\mathcal{B}}^* Q'(s, a)| \tag{25}$$

$$= \gamma \sup_{s,a} |\max\{h(s), \max_{s' \in \hat{T}(s,a)} \min_{a'} Q(s', a')\}$$

$$- \max\{h(s), \max_{s' \in \hat{T}(s,a)} \min_{a'} Q'(s', a')\}| \tag{26}$$

$$\le \gamma \sup_{s,a} |\max_{s' \in \hat{T}(s,a)} \min_{a'} Q(s', a') - \max_{s' \in \hat{T}(s,a)} \min_{a'} Q'(s', a')| \tag{27}$$

$$\le \gamma \sup_{s,a} \max_{s' \in \hat{T}(s,a)} \max_{a'} |Q(s', a') - Q'(s', a')| \tag{28}$$

$$\le \gamma \sup_{s',a'} |Q(s', a') - Q'(s', a')| \tag{29}$$

$$= \gamma \|Q - Q'\|_\infty, \tag{30}$$

where inequality 27 owns to inequality 24. Thus, $\bar{\mathcal{B}}^*$ is a $\gamma$ contraction mapping in the $\infty$-norm. Then, for any $Q, Q'$, if $Q \ge Q'$ pointwise, then

$$\begin{aligned}
\bar{\mathcal{B}}^* Q &= (1 - \gamma)h(s) + \gamma \max\{h(s), \max_{s' \in \hat{T}(s,a)} \min_{a'} Q(s', a')\} \\
&\ge (1 - \gamma)h(s) + \gamma \max\{h(s), \min_{a'} Q(s', a')\}, s' \sim T(s, a) \\
&\ge (1 - \gamma)h(s) + \gamma \max\{h(s), \min_{a'} Q'(s', a')\}, s' \sim T(s, a) \\
&= \mathcal{B}^* Q'.
\end{aligned} \tag{31}$$

Now, let $Q^0$ be any initial $Q$ function, and define $\bar{Q}^k = (\bar{\mathcal{B}}^*)^k Q_0, Q^k = (\mathcal{B}^*)^k Q_0$. Then, we have $\bar{Q}^k \ge Q^k, \forall k \in \mathbb{N}$. Therefore, taking the limits $\bar{Q}^*_h = \lim_{k \to \infty} \bar{Q}^k$ and $Q^*_{h,T} = \lim_{k \to \infty} Q^k$, we obtain $\bar{Q}^*_h(s, a) \ge Q^*_{h,T}(s, a)$ pointwise. $\qquad\square$

## B.2 PROOF OF THEOREM 4.6

**Lemma B.1.** *Suppose that Assumption 4.4 holds, then $\forall s \in \bar{S}_f^* = S - S_f^*$, it holds that $\forall a, Q_h^*(s, a) \geq h_{min} + \gamma^{H^*}(h_{max} - h_{min})$, where $Q_h^*$ is the learned action-value function by the feasible Bellman operator, and $h_{min}$ is the minimum value of $h(s)$, and $h_{max}$ is the value of $h(s)$ when it is unsafe, that is whenever $h(s) > 0$, $h(s) = h_{max}$.*

*Proof.* For any trajectory $\tau$ starting from the infeasible state $s_0$, suppose it first violates at horizon $t$, that is for $(s_0, a_0, s_1, a_1, \ldots, s_t)$, $h(s_t) = h_{\max}$. Therefore, we have

$$
\begin{aligned}
Q_h^*(s_t, a_t) &= (1 - \gamma)h(s_t) + \gamma \max\{h(s_t), \min_{a'} Q_h^*(s_{t+1}, a_{t+1})\} \\
&\geq (1 - \gamma)h_{\max} + \gamma h_{\max} = h_{\max}.
\end{aligned}
\tag{32}
$$

Similarly, by recursion we have

$$
\begin{aligned}
Q_h^*(s_{t-k}, a_{t-k}) &\geq (1 - \gamma)h_{\min}(1 + \gamma + \cdots + \gamma^{k-1}) + \gamma^k h_{\max} \\
&= h_{\min} + \gamma^k(h_{\max} - h_{\min}).
\end{aligned}
\tag{33}
$$

For that $\gamma < 1$ and $h_{\max} - h_{\min} > 0$, and the arbitrariness of trajectory $\tau$, we get

$$
\forall a_0, Q_h^*(s_0, a_0) \geq h_{\min} + \gamma^t(h_{\max} - h_{\min}) \geq h_{\min} + \gamma^{H^*}(h_{\max} - h_{\min}).
\tag{34}
$$

$\square$

Based on Lemma B.1, we can now proceed to the proof of Theorem 4.6.

*Proof.* For any infeasible state $s_0$, suppose $|\hat{T}| = N$, that is the ensemble dynamics model set has $N$ dynamics models, using the following rollout method

- Using each dynamics model in $\hat{T}$ to obtain the fist next state set $\{s\}_1$ with $N$ states.

- For each state in $\{s\}_1$, using each dynamics model to generate next rollout state set.

- Repeat the above two step, untill reaching horizon $H^*$.

This will generate $N^{H^*}$ rollout branches, and due to Assumption 4.4 and Assumption 4.5, there will be at least one branch that is unsafe. Therefore, if there is only one unsafe branch, then this will be the ground truth branch, and according to Lemma B.1, it can be ensured that $\forall a, Q_h^*(s_0, a) \geq h_{\min} + \gamma^{H^*}(h_{\max} - h_{\min})$. Otherwise, if there are more than one unsafe branch, then we don't need to figure out the ground truth branch, choosing any one of them to update the action value function can also lead to $\forall a, Q_h^*(s_0, a) \geq h_{\min} + \gamma^{H^*}(h_{\max} - h_{\min})$. Then, according to Proposition 4.3 and selecting $\gamma \in (\sqrt[H^*]{\frac{h_{\min}}{h_{\min} - h_{\max}}}, 1)$, we have

$$
\forall a, \bar{Q}_h^*(s_0, a) \geq Q_h^*(s_0, a) \geq h_{\min} + \gamma^{H^*}(h_{\max} - h_{\min}) > 0.
\tag{35}
$$

$\square$

## B.3 PROOF OF THEOREM 4.9

**Lemma B.2.** *(Janner et al., 2019) Suppose the expected KL-divergence between two transition distributions is bounded as $\max_t \mathbb{E}_{s \sim p_1^t(s)} D_{KL}(p_1(s'|s) || p_2(s'|s)) \leq \delta$, and the initial stat distributions are the same $p_1^0(s) = p_2^0(s)$. Then the distance in the state marginal is bounded as*

$$
D_{TV}(p_1^t(s) || p_2^t(s)) \leq t\delta.
\tag{36}
$$

*Proof.* Please refer to Janner et al. (2019). $\square$

Based on Lemma B.2, we can now proceed to the proof of Theorem 4.9.

*Proof.* First, according to Assumption 4.7, Assumption 4.8 and Lemma B.2, we have $|\mathbb{E}_{p_1^t}[h(s)] - \mathbb{E}_{p_2^t}[h(s)]| \leq tK\delta$, which means $\mathbb{E}_{p_2^t}[h(s)] \geq \mathbb{E}_{p_1^t}[h(s)] - tK\delta$ holds for any $t$. Meanwhile, similar to the proof of Lemma 1, it is clear that $Q_h^{p_2}(s,a) \geq h_{\min} + \gamma^t(\mathbb{E}_{p_2^t}[h(s)] - h_{\min})$, where $Q_h^{p_2}$ refers to the learned value function with the feasible Bellman operator and state transition distribution $p_2$. (A critical observation is $\forall a, \mathbb{E}_{s \sim p_2^t}[Q(s,a)] \geq \mathbb{E}_{s \sim p_2^t}[h(s)]$.)

Therefore, combining the results before and we can get $Q_h^{p_2}(s,a) \geq (1-\gamma^t)h_{\min} + \gamma^t \mathbb{E}_{p_1^t}[h(s)] - \gamma^t tK\delta$ holds for any $t$, which means

$$Q_h^{p_2}(s,a) \geq \max_{t \in \{1,\dots,H^*\}}[(1-\gamma^t)h_{\min} + \gamma^t \mathbb{E}_{p_1^t}[h(s)] - \gamma^t tK\delta]. \tag{37}$$

Finally, apply Proposition 4.3, we have

$$\bar{Q}_h^*(s,a) \geq Q_h^{p_2}(s,a) \geq \max_{t \in \{1,\dots,H^*\}}[(1-\gamma^t)h_{\min} + \gamma^t \mathbb{E}_{p_1^t}[h(s)] - \gamma^t tK\delta]$$
$$\geq (1-\gamma^{t^*})h_{\min} + \gamma^{t^*} \mathbb{E}_{p_1^{t^*}}[h(s)] - \gamma^{t^*} t^* K\delta, \tag{38}$$

where $t^* = \arg\max_{t \in \{1,\dots,H^*\}}[\mathbb{E}_{p_1^t}[h(s)]]$. □

### B.4 DERIVATION OF EQUATION (16)

First, owing to the closed-form solution of the optimal policy given in Equation 15, the policy optimization objective can be expressed as follows:

$$\max_{\pi_\theta} \mathbb{E}_{(s,a) \sim \mathcal{D}}[w(s,a) \log \pi_\theta(s,a)]. \tag{39}$$

When the policy is modeled using a diffusion model, it follows that

$$w(s,a) \log \pi_\theta(s,a) = w(s,a_0)\mathbb{E}_{a_{1:T} \sim q(a_{1:T}|s,a_0)}[\log \frac{\pi_\theta(a_{0:T}|s)}{q(a_{1:T}|s,a_0)}]$$
$$+ w(s,a_0)D_{\mathrm{KL}}(q(a_{1:T}|s,a_0)||\pi_\theta(a_{1:T}|s,a_0)), \tag{40}$$

where $a_0$ corresponds to the action $a$ from the offline dataset, and $q$ denotes the forward process of the diffusion model. Since $w(s,a) \geq 0$, it follows from Jensen's inequality that

$$w(s,a_0)D_{\mathrm{KL}}(q(a_{1:T}|s,a_0)||\pi_\theta(a_{1:T}|s,a_0))$$
$$= w(s,a_0)\mathbb{E}_{a_{1:T} \sim q(a_{1:T}|s,a_0)}[-\log \frac{\pi_\theta(a_{1:T}|s,a_0)}{q(a_{1:T}|s,a_0)}]$$
$$\geq -w(s,a_0)\log(\mathbb{E}_{a_{1:T} \sim q(a_{1:T}|s,a_0)}[\frac{\pi_\theta(a_{1:T}|s,a_0)}{q(a_{1:T}|s,a_0)}]) \tag{41}$$
$$= -w(s,a_0)\log 1$$
$$= 0.$$

Therefore, we have $\mathbb{E}_{(s,a_0) \sim \mathcal{D}}[w(s,a_0)\mathbb{E}_{a_{1:T} \sim q(a_{1:T}|s,a_0)}[\log \frac{\pi_\theta(a_{0:T}|s)}{q(a_{1:T}|s,a_0)}]]$ is the tight lower bound of $\mathbb{E}_{(s,a) \sim \mathcal{D}}[w(s,a)\log \pi_\theta(s,a)]$. The policy optimization objective can thus be written as

$$\max_{\pi_\theta} \mathbb{E}_{(s,a_0) \sim \mathcal{D}}[w(s,a_0)\mathbb{E}_{a_{1:T} \sim q(a_{1:T}|s,a_0)}[\log \frac{\pi_\theta(a_{0:T}|s)}{q(a_{1:T}|s,a_0)}]]. \tag{42}$$

For each state $s$, since

$$\mathbb{E}_{a_{1:T} \sim q(a_{1:T}|s,a_0)} \left[ \log \frac{q(a_{1:T}|a_0)}{\pi_\theta(a_{0:T})} \right]$$

$$= \mathbb{E}_{a_{1:T} \sim q(a_{1:T}|s,a_0)} \left[ \log \frac{\prod_{t=1}^T q(a_t|a_{t-1})}{\pi_\theta(a_T) \prod_{t=1}^T \pi_\theta(a_{t-1}|a_t)} \right]$$

$$= \mathbb{E}_{a_{1:T} \sim q(a_{1:T}|s,a_0)} \left[ -\log \pi_\theta(a_T) + \log \frac{\prod_{t=1}^T q(a_t|a_{t-1})}{\prod_{t=1}^T \pi_\theta(a_{t-1}|a_t)} \right]$$

$$= \mathbb{E}_{a_{1:T} \sim q(a_{1:T}|s,a_0)} \left[ -\log \pi_\theta(a_T) + \sum_{t=1}^T \log \frac{q(a_t|a_{t-1})}{\pi_\theta(a_{t-1}|a_t)} \right]$$

$$= \mathbb{E}_{a_{1:T} \sim q(a_{1:T}|s,a_0)} \left[ -\log \pi_\theta(a_T) + \sum_{t=2}^T \log \frac{q(a_t|a_{t-1})}{\pi_\theta(a_{t-1}|a_t)} + \log \frac{q(a_1|a_0)}{\pi_\theta(a_0|a_1)} \right]$$

$$= \mathbb{E}_{a_{1:T} \sim q(a_{1:T}|s,a_0)} \left[ -\log \pi_\theta(a_T) + \sum_{t=2}^T \log \frac{q(a_{t-1}|a_t,a_0)q(a_t|a_0)}{\pi_\theta(a_{t-1}|a_t)q(a_{t-1}|a_0)} + \log \frac{q(a_1|a_0)}{\pi_\theta(a_0|a_1)} \right]$$

$$= \mathbb{E}_{a_{1:T} \sim q(a_{1:T}|s,a_0)} \left[ -\log \pi_\theta(a_T) + \sum_{t=2}^T \left( \log \frac{q(a_{t-1}|a_t,a_0)}{\pi_\theta(a_{t-1}|a_t)} + \log \frac{q(a_t|a_0)}{q(a_{t-1}|a_0)} \right) + \log \frac{q(a_1|a_0)}{\pi_\theta(a_0|a_1)} \right]$$

$$= \mathbb{E}_{a_{1:T} \sim q(a_{1:T}|s,a_0)} \left[ -\log \pi_\theta(a_T) + \sum_{t=2}^T \log \frac{q(a_{t-1}|a_t,a_0)}{\pi_\theta(a_{t-1}|a_t)} + \log \frac{q(a_T|a_0)}{q(a_1|a_0)} + \log \frac{q(a_1|a_0)}{\pi_\theta(a_0|a_1)} \right]$$

$$= \mathbb{E}_{a_{1:T} \sim q(a_{1:T}|s,a_0)} \left[ -\log \pi_\theta(a_T) + \sum_{t=2}^T \log \frac{q(a_{t-1}|a_t,a_0)}{\pi_\theta(a_{t-1}|a_t)} + \log q(a_T|a_0) \right.$$

$$\left. -\log q(a_1|a_0) + \log q(a_1|a_0) - \log \pi_\theta(a_0|a_1) \right]$$

$$= \mathbb{E}_{a_{1:T} \sim q(a_{1:T}|s,a_0)} \left[ \log \frac{q(a_T|a_0)}{\pi_\theta(a_T)} + \sum_{t=2}^T \log \frac{q(a_{t-1}|a_t,a_0)}{\pi_\theta(a_{t-1}|a_t)} - \log \pi_\theta(a_0|a_1) \right]$$

$$= \mathbb{E}_{a_{1:T} \sim q(a_{1:T}|s,a_0)} \left[ D_{KL}\big(q(a_T|a_0)\|\pi_\theta(a_T)\big) + \sum_{t=2}^T D_{KL}\big(q(a_{t-1}|a_t,a_0)\|\pi_\theta(a_{t-1}|a_t)\big) - \log \pi_\theta(a_0|a_1) \right]$$

$$= \mathbb{E}_{a_0,\epsilon} \left[ \frac{\beta_t^2}{2\sigma_t^2 \alpha_t(1-\bar{\alpha}_t)} \left\| \epsilon - \epsilon_\theta\left(\sqrt{\bar{\alpha}_t}a_0 + \sqrt{1-\bar{\alpha}_t}\epsilon, s, t\right) \right\|^2 \right].$$

$$(43)$$

Thus the policy optimization objective can be obtained by

$$\max_{\pi_\theta} \mathbb{E}_{(s,a_0)\sim\mathcal{D}} \left[ w(s,a_0)\mathbb{E}_{a_{1:T}\sim q(a_{1:T}|s,a_0)}[\log \frac{\pi_\theta(a_{0:T}|s)}{q(a_{1:T}|s,a_0)}] \right]$$

$$\triangleq \min_{\pi_\theta} \mathbb{E}_{(s,a_0)\sim\mathcal{D}} \left[ w(s,a_0)\mathbb{E}_{a_{1:T}\sim q(a_{1:T}|s,a_0)}[\log \frac{q(a_{1:T}|s,a_0)}{\pi_\theta(a_{0:T}|s)}] \right] \qquad (44)$$

$$\triangleq \min_{\pi_\theta} \mathbb{E}_{(s,a)\sim\mathcal{D},\epsilon,t} \left[ \frac{\beta_t^2}{2\sigma_t^2 \alpha_t(1-\bar{\alpha}_t)} w(s,a)\|\epsilon - \epsilon_\theta(\sqrt{\bar{\alpha}_t}a + \sqrt{1-\bar{\alpha}_t}\epsilon, s, t)\|^2 \right].$$

In the above derivation, Equation 43 arises from the derivation of the DDPM (Ho et al., 2020) optimization objective, consistent with works such as QVPO (Ding et al., 2024).

## C  MORE RELATED WORK

**Offline RL.**  Offline RL trains policies using pre-collected datasets, avoiding real-world trial and error, which is critical for deploying RL in practical settings. Its primary challenge is addressing

the extrapolation error (Prudencio et al., 2023). Methods such as CQL (Kumar et al., 2020) and ICQ (Yang et al., 2021) penalize the value function of unseen actions in order to constrain actions to those observed in the offline dataset, while BCQ (Fujimoto et al., 2019) explicitly restricts candidate actions to remain close to the offline data distribution by leveraging a CVAE. Furthermore, approaches like IQL (Kostrikov et al., 2022) and DT (Chen et al., 2021) adopt BC-based supervised learning to fully confine the policy distribution within the offline dataset distribution, thereby avoiding the effect of extrapolation error. Others, such as MOReL (Kidambi et al., 2020), MOPO (Yu et al., 2020), and MOBILE (Sun et al., 2023), learn the environment models from the offline data and utilize these models with uncertainty estimates to avoid OOD regions with low model accuracy.

**Generated model assisted RL.** Driven by the powerful capacity of generative models to capture multimodal distributions, as well as their rapid progress in domains like natural language processing, recent years have witnessed a growing interest in applying such models to reinforcement learning (Li et al., 2025b). Apart from the aforementioned applications, LLMs can also serve other roles in RL, such as information representation and processing (Paischer et al., 2022), action space compression (Yan et al., 2025), world modeling (Ge et al., 2024), and multi-agent task allocation (Kannan et al., 2024). In addition to LLMs, other generative models are gaining growing traction in RL. Diffusion models, in particular, have emerged as the most widely applied due to their strong expressive power. They have been utilized for planning (Janner et al., 2022; Ajay et al., 2023), for offline policy learning (Wang et al., 2023; Chi et al., 2023; Ding & Jin, 2024), and more recently for online policy learning (Ding et al., 2024; Celik et al., 2025; Ma et al., 2025). Flow matching, due to its simpler generative process compared to diffusion models, has also recently attracted increasing attention in RL (Park et al., 2025; Espinosa-Dice et al., 2025; Lv et al., 2025).

## D  IMPLEMENTATION DETAILS

In this section, we will offer the implementation details of PROCO.

### D.1  MODEL LEARNING

We begin with model learning using the safe-only dataset $\mathcal{D}$. Specifically, this involves both dynamics model learning and reward model learning, which are jointly formulated as an ensemble dynamics model $\hat{T}$. Each model $T \in \hat{T}$ is formulated as a Gaussian distribution over the next state and reward given the current state and action:

$$T(s', r|s, a) = \mathcal{N}(\mu(s, a), \Sigma(s, a)). \tag{45}$$

Then, the ensemble dynamics model is optimized by

$$\max_{\hat{T}} \mathbb{E}_{(s,a,s',r)\sim\mathcal{D}}[\log \hat{T}(s', r|s, a)]. \tag{46}$$

In training, seven candidate models $T$ are trained to construct the ensemble dynamics model $\hat{T}$, and the best 5 models are picked based on the validation prediction error on a held-out set that contains 20% samples in $\mathcal{D}$. During model rollout, we randomly pick one dynamics model from the best 5 models to obtain $(s', r)$. For the cost label, given the cost function $\bar{c}$, to mitigate the impact of model uncertainty and obtain a more conservative estimate, we label a sample as 1 if any $s'$ of the five best models' outputs satisfies $\bar{c}(s') = 1$.

### D.2  CONSERVATIVE COST FUNCTION GENERATION

The key to LLM-based cost function generation is the formulation of the prompt input. Accordingly, this section details the prompt design employed in our approach.

First, we present the overall instruction used when initially employing the LLM for cost function generation

**Instruction**

You are an expert python coder. Please help me write some functions to evaluate whether a given observation is safe or not. The function should return 0 if the observation is safe, and 1 otherwise. The function should be called get_cost(observation).
The **cost description** is: {Cost Description}.
The **observation description** is: {Observation Description}.
Now, please give the function get_cost, you should be a little **more conservative**, some situations that do not violate safety constraints but are close to doing so can also be considered unsafe.

Subsequently, we present two examples to demonstrate the Cost Description, Observation Description within the instruction above, and Feedback provided for the generated cost function.

**Push2**  First, the Cost Description for the Push2 task is composed as follows:

**Push2 Cost Description**

If you are within 0.3 meters of hazard, it is unsafe. If you contact with a pillar (being very close to the pillar), it is unsafe.

The corresponding Observation Description is as follows:

**Push2 Observation Description**

The observation contains some agent specific dimensions plus 4 radar vetors, the last 16 dimensions is the push box radar, the 16 dimensions before push box radar is the pillar radar, the 16 dimensions before pillar radar is the hazard radar. Each radar has 16 dimensions, meaning taking the robot's forward direction as 0 degrees, determine whether a detected target is present at every 22.5-degree interval around it. However, there is no information about robot's forward direction in the observation. The max radar detection range is 3 meters. If in a given direction, the detected target is d meters away, and d≤3, then the radar will give the number as (3-d)/3. Meanwhile, since the observation lacks a collision sensor, contact with a target does not result in a radar reading of zero distance. This discrepancy is due to the physical volume of the target, whose radius generally does not exceed 0.3 meters.

After incorporating the above content into the instruction and inputting it into the LLM (GPT o4-mini), it first generated the following cost function:

**Push2 Output Function1**

```
def get_cost(observation):
    """
    Returns 0 if the observation is safe, 1 if unsafe.

    Parameters
    ----------
    observation : array-like, length >= 48
        The last 48 entries are three 16-dimensional radars:
          - hazard radar:  obs[-48:-32]
          - pillar radar:  obs[-32:-16]
          - push-box radar: obs[-16:]
        Radar values r correspond to distances d = 3*(1 - r).

    Safety rules (conservative):
      1. Any hazard reading r >= 0.90  →  unsafe
      2. Any pillar reading r >= 0.85  →  unsafe
```

```
    Returns
    -------
    int
        0 if safe, 1 if unsafe.
    """
    obs = np.asarray(observation)
    if obs.ndim != 1 or obs.size < 48:
        raise ValueError("Observation must be a 1D array with length
        ↪  >= 48.")

    # extract the 16-dim hazard and pillar readings
    hazard_r = obs[-48:-32]
    pillar_r = obs[-32:-16]

    # thresholds
    HAZARD_THRESH = 0.90   # exact 0.3 m
    PILLAR_THRESH = 0.85   # small margin beyond contact

    # check if any reading exceeds threshold
    too_close_to_hazard = np.any(hazard_r >= HAZARD_THRESH)
    contact_with_pillar = np.any(pillar_r >= PILLAR_THRESH)

    return 1 if (too_close_to_hazard or contact_with_pillar) else 0
```

However, upon evaluation, this cost function labeled 2% of the samples in $\mathcal{D}$ as unsafe, which does not meet our conservative requirement of labeling samples within $[p_{\min}, p_{\max}]$ as unsafe, where $p_{\min} = 10\%$ and $p_{\max} = 30\%$ in practice. Therefore, we provided the following feedback:

---

**Push2 Feedback1**

For safe samples, this function classifies 2% of them as unsafe. We want to classify 10%-30% safe samples as unsafe. It should be a little more conservative.

---

Based on the above feedback, the LLM made corresponding modifications to the output function

---

**Push2 Output Function2**

```
def get_cost(observation):
    """
    Returns 0 if the observation is safe, 1 if unsafe
    ↪  (conservatively).

    Parameters
    ----------
    observation : array-like, length >= 48
        The last 48 entries are:
            - hazard radar:  obs[-48:-32]
            - pillar radar:  obs[-32:-16]
            - push-box radar: obs[-16:]

    Conservative thresholds:
      - Hazard: r >= 0.80 → d <= 0.60 m (unsafe)
      - Pillar: r >= 0.75 → d <= 0.75 m (unsafe)

    Returns
    -------
    int
        0 if safe, 1 if unsafe.
    """
    obs = np.asarray(observation)
    if obs.ndim != 1 or obs.size < 48:
```

```
        raise ValueError("Observation must be a 1D array with length
        ↪  >= 48.")

    hazard_r = obs[-48:-32]
    pillar_r = obs[-32:-16]

    HAZARD_THRESH = 0.80   # ~0.60 m
    PILLAR_THRESH = 0.75   # ~0.75 m

    too_close_to_hazard = np.any(hazard_r >= HAZARD_THRESH)
    close_to_pillar = np.any(pillar_r >= PILLAR_THRESH)

    return 1 if (too_close_to_hazard or close_to_pillar) else 0
```

Nevertheless, the revised cost function proved excessively conservative, labeling 40% of safe samples as unsafe. Consequently, we supplied the LLM with additional feedback:

**Push2 Feedback2**

For safe samples, this function classifies 40% of them as unsafe. We want to classify 10%-30% safe samples as unsafe. It is too conservative.

Finally, the LLM adjusted the cost function once again, yielding:

**Push2 Output Function3**

```
def get_cost(observation):
    """
    Returns 0 if the observation is safe, 1 if unsafe (moderately
    ↪  conservative).

    Parameters
    ----------
    observation : array-like, length >= 48
        The last 48 entries are:
            - hazard radar:  obs[-48:-32]
            - pillar radar:  obs[-32:-16]
            - push-box radar: obs[-16:]

    Thresholds tuned for 10-30% false positives on safe samples:
      - Hazard: r >= 0.85 → d <= 0.45 m (unsafe)
      - Pillar: r >= 0.80 → d <= 0.60 m (unsafe)

    Returns
    -------
    int
        0 if safe, 1 if unsafe.
    """
    obs = np.asarray(observation)
    if obs.ndim != 1 or obs.size < 48:
        raise ValueError("Observation must be a 1D array with length
        ↪  >= 48.")

    hazard_r = obs[-48:-32]
    pillar_r = obs[-32:-16]

    HAZARD_THRESH = 0.85   # ~0.45 m
    PILLAR_THRESH = 0.80   # ~0.60 m

    too_close_to_hazard = np.any(hazard_r >= HAZARD_THRESH)
    close_to_pillar = np.any(pillar_r >= PILLAR_THRESH)
```

```
        return 1 if (too_close_to_hazard or close_to_pillar) else 0
```

**Walker2dVelocity** In MuJoCo Velocity tasks, the agent is deemed unsafe once its instantaneous velocity surpasses a predefined threshold. Nevertheless, the velocity included in the agent's observation reflects only the terminal velocity at the end of a step, which may deviate from the velocity during step execution. Consequently, an LLM-generated cost function may align well with the textual description, or even adopt a more conservative stance, but still fail to achieve perfect classification accuracy on the 100 samples in $\mathcal{D}_{\text{unsafe}}$. This example illustrates such a case. First, the Cost Description for the Walker2dVelocity task is composed as follows:

---

**Walker2dVelocity Cost Description**

If the velocity of the robot exeeds 2.3415, it is unsafe. When calculating the velocity, we only consider x-coordinate and y-coordinate (if exists).

---

The corresponding Observation Description is as follows:

---

**Walker2dVelocity Observation Description**

The observation contains 17 dimensions. The first is z-coordinate of the front tip; the second is angle of the torso; the third is angle of the thigh joint; the forth is angle of the leg joint; the fifth is angle of the foot joint; the sixth is angle of the left thigh joint; the seventh is angle of the left leg joint; the eighth is angle of the left foot joint; the nineth is velocity of the x-coordinate of the torso; the tenth is velocity of the z-coordinate (height) of the torso; the eleventh is angular velocity of the angle of the torso; the twelveth is angular velocity of the thigh hinge; the thirteenth is angular velocity of the leg hinge; the forteenth is angular velocity of the foot hinge; the fifteenth is angular velocity of the thigh hinge; the sixteenth is angular velocity of the leg hinge; the seventeenth is angular velocity of the foot hinge.

---

After incorporating the above content into the instruction and inputting it into the LLM, it first generated the following cost function:

---

**Walker2dVelocity Output Function1**

```python
def get_cost(observation, max_safe_vel=2.3415, safety_margin=0.05):
    """
    Return 0 if the observation is safe, 1 otherwise.

    We consider only the torso's x-velocity (observation[8]).
    Any |velocity| >= (1 - safety_margin) * max_safe_vel is deemed
    ↪  unsafe.

    Parameters
    ----------
    observation : sequence of float, length >= 9
        The 17-dimensional observation vector.
    max_safe_vel : float
        The hard safety limit on velocity.
    safety_margin : float
        Fractional margin to be conservative (e.g. 0.05 means 5%
        ↪  below max_safe_vel).

    Returns
    -------
    int
        0 if safe, 1 if unsafe.
```

---

```
    """
    # pick out the x-velocity of the torso
    v_x = observation[8]

    # define the conservative threshold
    threshold = (1.0 - safety_margin) * max_safe_vel

    # unsafe if we're at or above the threshold
    return 1 if abs(v_x) >= threshold else 0
```

However, the accuracy of this function on $\mathcal{D}_{\text{unsafe}}$ is only 98%. Therefore, we provided feedback to the LLM:

**Walker2dVelocity Feedback1**

For 100 unsafe testing samples, this function achieves 98% accuracy, it should be a little more conservative.

Based on the above feedback, the LLM made corresponding modifications to the output function

**Walker2dVelocity Output Function2**

```
def get_cost(observation, max_safe_vel=2.3415, safety_margin=0.10):
    """
    Return 0 if the observation is safe, 1 otherwise.

    Unsafe if torso's x-velocity exceeds (1 - safety_margin) *
    ↪  max_safe_vel.
    This version is more conservative, using a 10% margin.

    Parameters
    ----------
    observation : sequence of float, length >= 9
        The 17-dimensional observation vector.
    max_safe_vel : float
        The hard velocity safety threshold.
    safety_margin : float
        A conservative margin to catch near-threshold values.

    Returns
    -------
    int
        0 if safe, 1 if unsafe.
    """
    v_x = observation[8]
    threshold = (1.0 - safety_margin) * max_safe_vel
    return 1 if abs(v_x) >= threshold else 0
```

The accuracy of this function on $\mathcal{D}_{\text{unsafe}}$ remains at only 99%. Therefore, we provided additional feedback

**Walker2dVelocity Feedback2**

For 100 unsafe testing samples, this function achieves 99% accuracy, it should be a little more conservative.

Finally, the LLM adjusted the cost function once again, yielding:

**Walker2dVelocity Output Function3**

```python
def get_cost(observation, max_safe_vel=2.3415, safety_margin=0.15):
    """
    Return 0 if the observation is safe, 1 otherwise.

    Unsafe if torso's x-velocity magnitude exceeds
    (1 - safety_margin) * max_safe_vel.

    This default uses a 15% margin (i.e. flags |v_x| >= ~1.9903).

    Parameters
    ----------
    observation : sequence of float, len >= 9
        The 17-dimensional observation vector.
    max_safe_vel : float
        The hard velocity safety limit.
    safety_margin : float
        Fractional buffer under max_safe_vel (default 0.15).

    Returns
    -------
    int
        0 if safe, 1 if unsafe.
    """
    # torso x-velocity is at index 8
    v_x = observation[8]

    # build a conservative threshold
    threshold = (1.0 - safety_margin) * max_safe_vel

    return 1 if abs(v_x) >= threshold else 0
```

## D.3 POLICY LEARNING

During model rollouts, we follow the rollout procedure used in works such as MOPO, performing a rollout after every $K$ policy gradient updates. For each rollout, a batch of size $b$ is sampled from $\mathcal{D}$, and each data point in the batch is rolled out for $H$ steps. The process of sampling and branch rollout is repeated $E$ times. In practice, we set $K = 250000$, $b = 50000$, $H = 1$, and $E = 10$.

The process of policy learning remains largely consistent with the description in the main text, where Equation (12) is used to learn constraint violation value functions, and Equation (13) is employed to learn reward value functions. During policy extraction, the optimal policy is represented as

$$\pi^*(a|s) = w(s,a)\pi_\beta(a|s), \ w(s,a) = \begin{cases} \exp(\alpha_1 A_r^*(s,a)) \cdot \mathbb{I}_{Q_h^*(s,a) \leq 0} & V_h^*(s) \leq 0 \\ \exp(-\alpha_2 A_h^*(s,a)) & V_h^*(s) > 0 \end{cases}, \quad (47)$$

where the normalization terms in $w(s,a)$ are omitted. Finally, the optimization is performed as follows:

$$\mathcal{L}_\theta = \mathbb{E}_{(s,a)\sim\mathcal{D},\epsilon,t} \left[ w(s,a)||\epsilon - \epsilon_\theta(\sqrt{\bar{\alpha}_t}a + \sqrt{1 - \bar{\alpha}_t}\epsilon, s, t)||^2 \right], \quad (48)$$

where the weighting term $\frac{\beta_t^2}{2\sigma_t^2\alpha_t(1-\bar{\alpha}_t)}$ in Equation (16) is omitted.

Finally, the detailed pseudo-code for PROCO is provided in Algorithm 1.

## D.4 HYPERPARAMETERS

The training of PROCO involves the selection of hyperparameters. To ensure reproducibility, this section outlines the specific hyperparameters used in our experiments, as shown in Table 2. Specifically, for the expectile parameter $\tau$, we set it to $0.9$ in the Point and Car tasks, and to $0.95$ in the MuJoCo Velocity tasks. The diffusion action candidates hyperparameter denotes the procedure at test time in which the diffusion policy samples 16 candidate actions and selects the action with the

---

**Algorithm 1 PROCO**

---

**Input:** offline dataset $\mathcal{D}$, LLM generated conservative cost function $\bar{c}$.
**Initialize:** ensemble model $\hat{T}$, constraint violation value networks $Q_h, V_h$, reward value networks $Q_r, V_r$, diffusion policy $\epsilon_\theta$, rollout dataset $\mathcal{D}_{\hat{T}} = \emptyset$.
**for** step in dynamics model training steps **do**
    Update $\hat{T}$ with Equation (46).
**end for**
**for** step in policy training steps **do**
    **if** step % rollout frequency == 0 **then**
        Obtain rollout trajectories $\{\tau_i\}_{i=1}^N$ and add them to $\mathcal{D}_{\hat{T}}$ according to Section 4.3.
    **end if**
    Update $V_h, Q_h$ with Equation (12).
    Update $V_r, Q_r$ with Equation (13).
    Update $\epsilon_\theta$ with Equation (48)
**end for**
Return $V_h, Q_h, V_r, Q_r, \epsilon_\theta$.

---

minimum constraint violation value function, corresponding to the highest level of safety. PROCO is implemented based on OSRL [1] and FISOR [2] code bases, and the default parameters are retained for any hyperparameters not explicitly mentioned.

Table 2: Hyperparameter choices of PROCO.

| | Hyperparameter | Value |
|---|---|---|
| model learning | model hidden layers | $[256, 256, 256, 256]$ |
| | layer weight decays | $[2.5e-5, 5e-5, 7.5e-5, 7.5e-5, 1e-4]$ |
| | model ensemble number | 7 |
| | model elites number | 5 |
| | batch size | 512 |
| | learning rate | 0.001 |
| cost function generation | $p_{\min}$ | 10% |
| | $p_{\max}$ | 30% |
| | max LLM query number | 10 |
| policy learning | network hidden layers | $[256, 256]$ |
| | model rollout frequency | $2.5e5$ |
| | model rollout batch size | 50000 |
| | model rollout length | 1 |
| | model rollout epoch | 10 |
| | model rollout std | 0.1 |
| | expectile $\tau$ | $\{0.9, 0.95\}$ |
| | $\alpha_1$ | 3 |
| | $\alpha_2$ | 5 |
| | exponential advantage clip (feasible) | $(-\infty, 100]$ |
| | exponential advantage clip (infeasible) | $(-\infty, 150]$ |
| | diffusion step | 5 |
| | diffusion action candidates | 16 |
| | value batch size | 256 |
| | diffusion batch size | 2048 |
| | learning rate | $3e-4$ |
| | $\gamma$ | 0.99 |
| | soft update $\alpha$ | 0.001 |
| | training steps | $2e6$ |

---

[1]https://github.com/liuzuxin/OSRL
[2]https://github.com/ZhengYinan-AIR/FISOR

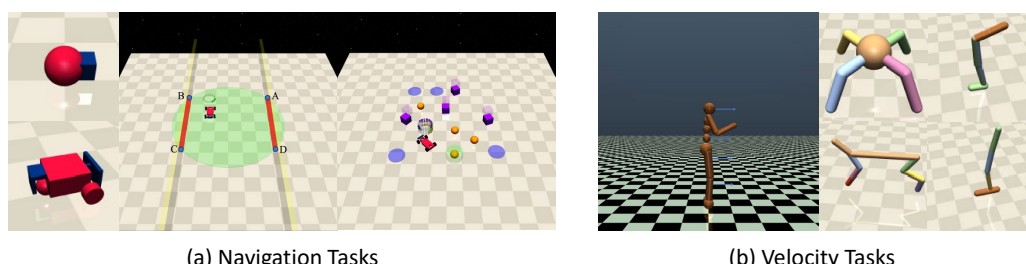

(a) Navigation Tasks
(b) Velocity Tasks

Figure 5: Tasks used in this paper. (a) Navigation Tasks based on Point and Car robots. (b) Velocity Tasks based on Ant, HalfCheetah, Hopper, Walker2d, and Swimmer robots.

# E DETAILED DESCRIPTION OF THE TASKS AND BASELINES

## E.1 TASKS

All tasks used in this paper are derived from the Safety-Gymnasium's Navigation Tasks and Velocity Tasks. In the Navigation Tasks, there are two different types of robots: Point and Car, which we need to control to navigate through the environment and earn rewards by reaching target points, pressing the correct buttons, or moving in designated directions. Different tasks also have varying costs, such as avoiding collisions with specific targets, preventing incorrect button presses, and staying within designated boundaries.

The Velocity Tasks are built on traditional MuJoCo (Todorov et al., 2012) simulations, requiring robots such as Ant, HalfCheetah, Swimmer, and Walker2d to move, where higher speeds result in higher rewards. However, each robot has specific safety velocity thresholds for different tasks, and exceeding these thresholds leads to unsafe states. For detailed descriptions of each task, refer to the original Safety-Gymnasium paper (Ji et al., 2023).

## E.2 BASELINES

We provide a more detailed introduction to the baselines in this section.

- **CPQ** is a CMDP-based offline safe RL algorithm built upon the classic offline RL method CQL. It incorporates the conservative regularization operator from CQL into the cost critic, treating out-of-distribution samples as unsafe. Unlike traditional methods that use Lagrangian multipliers for policy updates, CPQ directly truncates the reward critic to 0 for unsafe state-action pairs, preventing unsafe policy execution.

- **BCQ Lagrange**, similar to CPQ, applies the Lagrangian multiplier method from safe RL to the classical offline RL algorithm BCQ. It learns a CVAE to fit the action distribution in the offline dataset, thereby ensuring that the candidate action set during policy learning remains as close as possible to the offline data distribution.

- **CDT** is the previous SOTA algorithm under sequence modeling for offline safe RL. After incorporating CTG into DT, CDT seeks to address the conflict between safety constraints and reward maximization. To tackle this issue, CDT proposes a data augmentation approach, where reward returns for certain safe but low-reward trajectories in the offline dataset are re-labeled with higher values.

- **COptiDICE** is the first to apply the advantage-weighted BC-based DICE algorithm to offline safe RL. By introducing Lagrangian multipliers, it combines the reward advantage and cost as the weighting term in BC, enabling efficient safe policy learning under soft constraints.

- **FISOR** is the SOTA offline safe RL algorithm based on hard constraint modeling. Similar to other algorithms under hard constraints Yu et al. (2022); Ganai et al. (2023), it first divides the state space into feasible and infeasible regions. It then uses the IQL Kostrikov et al. (2022) algorithm to learn feasible value functions for offline feasible region identification. Next, FISOR sets distinct learning objectives for the feasible and infeasible regions. In

## F  MORE EXPERIMENTAL RESULTS

### F.1  INTEGRATE PROCO WITH OTHER ALGORITHMS

Although in this work, PROCO leverages the unsafe samples generated from the dynamics model and conservative cost function for feasibility identification, and adopts the policy optimization scheme of FISOR under hard constraint modeling for policy extraction, the framework of proactive unsafe sample generation via the conservative cost function and dynamics model can in fact be flexibly combined with various value-based offline safe RL methods for minimal cost safe policy learning. Therefore, in this section, we further evaluate PROCO in combination with three value-based offline safe RL baselines under soft constraint modeling, with the goal of promoting more effective cost value function learning when no unsafe samples are available. As shown in Table 3, all three baselines achieve significant improvements in safety after being integrated with PROCO. These results further validate the effectiveness of PROCO's proactive unsafe sample generation framework for offline safe policy learning in scenarios with few or no unsafe samples.

Table 3: Results of integrate PROCO with other algorithms.

| Method | CarButton1 | | CarGoal1 | | CarPush1 | | Average | |
|---|---|---|---|---|---|---|---|---|
| | r↑ | c↓ | r↑ | c↓ | r↑ | c↓ | r↑ | c↓ |
| CPQ | $0.35 \pm 0.05$ | $42.38 \pm 0.81$ | $0.80 \pm 0.04$ | $5.74 \pm 1.74$ | $-0.01 \pm 0.27$ | $5.08 \pm 3.22$ | 0.38 | 17.73 |
| CPQ+PROCO | **0.06±0.05** | **7.08±1.67** | **0.46±0.13** | **3.04±0.52** | **-0.13±0.17** | **1.76±0.96** | **0.13** | **3.96** |
| BCQ Lagrange | $-0.02 \pm 0.04$ | $5.56 \pm 0.29$ | $0.47 \pm 0.02$ | $2.86 \pm 0.60$ | $0.20 \pm 0.02$ | $1.59 \pm 0.33$ | 0.22 | 3.34 |
| BCQ Lagrange+PROCO | **-0.09±0.04** | **2.42±0.21** | **0.35±0.00** | **2.36±0.61** | **0.19±0.02** | **1.02±0.26** | **0.15** | **1.93** |
| COptiDICE | $-0.04 \pm 0.08$ | $3.69 \pm 1.07$ | $0.38 \pm 0.07$ | $2.14 \pm 0.58$ | $0.20 \pm 0.02$ | $2.21 \pm 1.07$ | 0.18 | 2.68 |
| COptiDICE+PROCO | **-0.14±0.04** | **1.85±0.67** | **0.31±0.10** | **1.73±0.59** | **0.22±0.03** | **1.76±0.42** | **0.13** | **1.78** |

Table 4: Detailed standard deviation results of the main experiment. All results are computed using three different random seeds.

| Task | CPQ | | BCQ Lagrange | | BC | | CDT | | COptiDICE | | FISOR | | PROCO | |
|---|---|---|---|---|---|---|---|---|---|---|---|---|---|---|
| | r | c | r | c | r | c | r | c | r | c | r | c | r | c |
| PointButton1 | 0.05 | 0.64 | 0.08 | 0.80 | 0.05 | 1.21 | 0.14 | 1.90 | 0.02 | 0.68 | 0.12 | 0.91 | 0.01 | 0.56 |
| PointButton2 | 0.03 | 1.90 | 0.07 | 3.18 | 0.08 | 1.57 | 0.02 | 2.28 | 0.04 | 0.69 | 0.06 | 0.41 | 0.02 | 0.88 |
| PointGoal1 | 0.02 | 0.75 | 0.05 | 1.01 | 0.03 | 0.13 | 0.01 | 0.55 | 0.09 | 0.52 | 0.01 | 0.52 | 0.00 | 0.44 |
| PointGoal2 | 0.06 | 2.26 | 0.06 | 0.37 | 0.05 | 2.15 | 0.01 | 0.60 | 0.09 | 0.41 | 0.03 | 1.01 | 0.05 | 0.43 |
| PointPush1 | 0.05 | 4.47 | 0.03 | 0.97 | 0.03 | 0.91 | 0.03 | 0.15 | 0.02 | 1.36 | 0.06 | 1.27 | 0.00 | 0.53 |
| PointPush2 | 0.12 | 4.54 | 0.02 | 0.74 | 0.06 | 1.04 | 0.02 | 0.89 | 0.04 | 1.01 | 0.06 | 1.79 | 0.02 | 0.18 |
| CarButton1 | 0.05 | 0.81 | 0.04 | 0.29 | 0.01 | 1.35 | 0.02 | 1.86 | 0.08 | 1.07 | 0.03 | 1.78 | 0.04 | 0.23 |
| CarButton2 | 0.04 | 2.13 | 0.02 | 1.48 | 0.10 | 1.18 | 0.03 | 1.15 | 0.08 | 0.72 | 0.04 | 0.94 | 0.00 | 0.21 |
| CarGoal1 | 0.04 | 1.74 | 0.02 | 0.60 | 0.01 | 0.56 | 0.04 | 0.15 | 0.07 | 0.58 | 0.01 | 0.91 | 0.02 | 0.03 |
| CarGoal2 | 0.00 | 3.58 | 0.04 | 0.22 | 0.03 | 0.38 | 0.01 | 0.38 | 0.07 | 0.56 | 0.02 | 0.20 | 0.02 | 0.50 |
| CarPush1 | 0.27 | 3.22 | 0.02 | 0.33 | 0.05 | 0.24 | 0.01 | 1.13 | 0.02 | 1.07 | 0.04 | 0.35 | 0.02 | 0.86 |
| CarPush2 | 0.04 | 8.84 | 0.03 | 2.51 | 0.03 | 1.21 | 0.01 | 1.43 | 0.03 | 1.00 | 0.06 | 1.84 | 0.00 | 0.06 |
| SwimmerVelocityV1 | 0.14 | 11.09 | 0.10 | 17.19 | 0.10 | 5.05 | 0.02 | 3.13 | 0.06 | 2.37 | 0.01 | 0.56 | 0.02 | 0.12 |
| HopperVelocityV1 | 0.03 | 2.32 | 0.16 | 4.27 | 0.17 | 0.97 | 0.03 | 0.66 | 0.01 | 1.67 | 0.01 | 0.15 | 0.01 | 0.10 |
| HalfCheetahVelocityV1 | 0.08 | 4.63 | 0.01 | 11.22 | 0.00 | 2.94 | 0.01 | 1.01 | 0.01 | 0.00 | 0.01 | 3.45 | 0.01 | 0.00 |
| Walker2dVelocityV1 | 0.02 | 0.73 | 0.01 | 0.16 | 0.09 | 1.17 | 0.02 | 0.60 | 0.01 | 0.86 | 0.03 | 0.95 | 0.03 | 0.09 |
| AntVelocityV1 | 0.00 | 0.00 | 0.16 | 10.59 | 0.01 | 3.91 | 0.00 | 0.31 | 0.00 | 0.64 | 0.00 | 0.76 | 0.04 | 0.00 |
| Average | 0.06 | 3.16 | 0.05 | 3.29 | 0.05 | 1.53 | 0.03 | 1.07 | 0.04 | 0.89 | 0.04 | 1.05 | 0.02 | 0.31 |

## F.2 STANDARD DEVIATION OF THE MAIN RESULTS

In this section, we present the standard deviation of PROCO and various baselines computed with three different random seeds, as shown in Table 4. The results demonstrate that PROCO achieves lower standard deviation compared to other baselines, indicating its stability.

## F.3 DETAILED ABLATION RESULTS

In this section, we also provide the detailed results of the ablation studies, as shown in Table 5.

Table 5: Detailed ablation results.

| Task | W/o Model | | Full Model | | W/o Relabel | |
|---|---|---|---|---|---|---|
| | r↑ | c↓ | r↑ | c↓ | r↑ | c↓ |
| PointButton1 | 0.06±0.00 | 2.84±0.58 | 0.01±0.00 | 1.82±0.76 | 0.07±0.01 | 1.10±0.39 |
| PointButton2 | 0.09±0.03 | 4.32±0.28 | 0.03±0.02 | 2.14±0.72 | 0.09±0.02 | 2.19±0.36 |
| PointGoal1 | 0.54±0.04 | 2.41±0.76 | 0.07±0.02 | 0.18±0.13 | 0.28±0.04 | 0.25±0.13 |
| PointGoal2 | 0.14±0.03 | 1.58±0.68 | 0.03±0.01 | 0.63±0.34 | 0.09±0.05 | 0.92±0.30 |
| PointPush1 | 0.25±0.02 | 1.03±0.32 | 0.13±0.04 | 0.79±0.68 | 0.21±0.02 | 0.60±0.32 |
| PointPush2 | 0.11±0.03 | 1.21±0.36 | 0.04±0.05 | 0.48±0.25 | 0.07±0.03 | 1.22±0.27 |
| CarButton1 | -0.02±0.01 | 1.24±0.25 | 0.00±0.00 | 1.57±1.22 | -0.03±0.03 | 1.22±0.37 |
| CarButton2 | 0.00±0.02 | 3.22±0.94 | -0.06±0.01 | 1.99±0.80 | -0.03±0.03 | 1.76±1.04 |
| CarGoal1 | 0.27±0.02 | 0.37±0.13 | 0.02±0.02 | 0.00±0.00 | 0.15±0.06 | 0.31±0.18 |
| CarGoal2 | 0.06±0.03 | 0.14±0.12 | -0.01±0.02 | 0.07±0.07 | 0.11±0.02 | 1.00±0.34 |
| CarPush1 | 0.24±0.04 | 1.03±0.69 | 0.16±0.03 | 0.12±0.17 | 0.17±0.02 | 0.29±0.19 |
| CarPush2 | 0.03±0.03 | 0.48±0.03 | -0.03±0.05 | 0.00±0.00 | 0.06±0.02 | 1.27±0.53 |
| SwimmerVelocityV1 | 0.01±0.01 | 0.00±0.00 | 0.02±0.00 | 0.00±0.00 | 0.01±0.02 | 0.07±0.10 |
| HopperVelocityV1 | 0.17±0.05 | 0.17±0.12 | 0.04±0.03 | 0.00±0.00 | 0.12±0.09 | 0.18±0.20 |
| HalfCheetahVelocityV1 | 0.51±0.04 | 0.00±0.00 | 0.23±0.05 | 0.00±0.00 | 0.83±0.04 | 1.40±1.89 |
| Walker2dVelocityV1 | 0.15±0.03 | 1.47±0.50 | 0.05±0.03 | 0.01±0.01 | 0.18±0.05 | 3.36±1.38 |
| AntVelocityV1 | 0.59±0.01 | 0.00±0.00 | 0.00±0.08 | 0.01±0.01 | 0.52±0.05 | 0.01±0.01 |
| Average | 0.19 | 1.27 | 0.05 | 0.58 | 0.17 | 1.01 |

| Task | Det. Rollout | | W/o Consv. | | W/o Refl. | |
|---|---|---|---|---|---|---|
| | r↑ | c↓ | r↑ | c↓ | r↑ | c↓ |
| PointButton1 | 0.02±0.04 | 1.67±0.97 | 0.09±0.03 | 4.01±0.27 | 0.06±0.03 | 3.40±1.70 |
| PointButton2 | 0.07±0.01 | 2.90±0.64 | 0.08±0.01 | 3.54±0.80 | 0.07±0.01 | 4.55±1.48 |
| PointGoal1 | 0.40±0.02 | 1.65±0.36 | 0.37±0.00 | 0.63±0.21 | 0.34±0.04 | 0.86±0.55 |
| PointGoal2 | 0.06±0.01 | 0.25±0.18 | 0.06±0.01 | 0.74±0.54 | 0.08±0.02 | 0.42±0.13 |
| PointPush1 | 0.18±0.04 | 0.91±1.10 | 0.22±0.02 | 0.94±0.14 | 0.20±0.05 | 2.17±2.38 |
| PointPush2 | 0.08±0.04 | 0.51±0.38 | 0.11±0.04 | 3.98±1.95 | 0.07±0.01 | 1.22±0.90 |
| CarButton1 | -0.04±0.04 | 1.75±0.59 | -0.04±0.02 | 2.76±1.10 | -0.01±0.03 | 2.57±1.32 |
| CarButton2 | -0.01±0.01 | 1.36±0.27 | -0.01±0.00 | 2.40±0.72 | 0.00±0.01 | 3.21±1.44 |
| CarGoal1 | 0.15±0.04 | 0.03±0.04 | 0.23±0.04 | 0.50±0.37 | 0.16±0.02 | 0.16±0.14 |
| CarGoal2 | 0.03±0.00 | 0.08±0.11 | 0.03±0.00 | 0.39±0.49 | 0.04±0.03 | 0.80±0.93 |
| CarPush1 | 0.18±0.03 | 1.87±1.23 | 0.22±0.03 | 0.45±0.24 | 0.11±0.04 | 1.86±2.43 |
| CarPush2 | 0.00±0.00 | 0.04±0.05 | 0.04±0.03 | 3.00±1.34 | 0.03±0.04 | 1.19±0.97 |
| SwimmerVelocityV1 | 0.00±0.01 | 0.03±0.02 | 0.01±0.01 | 0.00±0.00 | 0.00±0.00 | 0.23±0.33 |
| HopperVelocityV1 | 0.20±0.12 | 0.01±0.01 | 0.06±0.08 | 0.00±0.00 | 0.13±0.17 | 0.00±0.00 |
| HalfCheetahVelocityV1 | 0.49±0.02 | 0.00±0.00 | 0.81±0.01 | 0.00±0.00 | 0.65±0.07 | 0.00±0.00 |
| Walker2dVelocityV1 | 0.15±0.01 | 1.57±0.36 | 0.22±0.04 | 2.28±0.15 | 0.18±0.00 | 2.09±0.57 |
| AntVelocityV1 | 0.46±0.05 | 0.00±0.00 | 0.41±0.03 | 0.00±0.00 | 0.50±0.18 | 0.00±0.00 |
| Average | 0.14 | 0.86 | 0.17 | 1.51 | 0.15 | 1.45 |

## F.4 DETAILED SENSITIVITY ANALYSIS RESULTS

Finally, in this section, we provide the detailed results of sensitivity studies, as shown in Table 6. Notably, while safety performance degrades significantly when $H = 5$, this degradation is mainly observed in the Point and Car tasks, whereas the MuJoCo Velocity tasks do not exhibit such a decline and may even become more conservative. These results suggest that when the environment model is sufficiently accurate, increasing $H$ can indeed yield safety performance; however, if the environment model accuracy is low, keeping $H$ small is preferable to avoid the negative impact of model errors on learning stability.

Table 6: Detailed sensitivity analysis results.

| Task | $E = 2$ | | $E = 5$ | | $H = 2$ | |
|---|---|---|---|---|---|---|
| | r↑ | c↓ | r↑ | c↓ | r↑ | c↓ |
| PointButton1 | 0.03±0.01 | 1.93±0.14 | 0.03±0.00 | 1.55±0.55 | -0.02±0.01 | 0.57±0.20 |
| PointButton2 | 0.06±0.04 | 2.31±1.15 | 0.06±0.01 | 2.42±0.58 | -0.06±0.05 | 1.79±0.33 |
| PointGoal1 | 0.45±0.03 | 1.70±1.03 | 0.42±0.07 | 1.28±0.66 | 0.14±0.03 | 2.67±0.31 |
| PointGoal2 | 0.10±0.04 | 0.71±0.45 | 0.08±0.02 | 0.53±0.31 | -0.04±0.04 | 1.20±0.27 |
| PointPush1 | 0.20±0.02 | 0.43±1.35 | 0.18±0.02 | 1.14±0.53 | 0.13±0.05 | 1.10±0.44 |
| PointPush2 | 0.10±0.00 | 0.76±0.31 | 0.09±0.02 | 0.24±0.16 | 0.06±0.02 | 0.65±0.25 |
| CarButton1 | -0.04±0.05 | 0.61±0.42 | -0.07±0.05 | 0.31±0.11 | -0.04±0.03 | 0.94±0.04 |
| CarButton2 | -0.02±0.04 | 1.85±0.29 | -0.03±0.03 | 1.68±0.43 | -0.13±0.06 | 0.75±0.27 |
| CarGoal1 | 0.22±0.02 | 0.15±0.10 | 0.20±0.02 | 0.03±0.05 | 0.16±0.08 | 1.45±0.59 |
| CarGoal2 | 0.05±0.02 | 0.58±0.26 | 0.05±0.00 | 0.77±0.11 | 0.05±0.04 | 2.29±0.89 |
| CarPush1 | 0.19±0.04 | 0.35±0.22 | 0.17±0.03 | 0.45±0.39 | 0.12±0.02 | 0.32±0.09 |
| CarPush2 | 0.03±0.01 | 0.01±0.01 | -0.01±0.02 | 0.00±0.00 | -0.01±0.04 | 0.64±0.91 |
| SwimmerVelocityV1 | 0.01±0.00 | 0.11±0.15 | 0.02±0.01 | 0.32±0.45 | 0.01±0.01 | 0.00±0.00 |
| HopperVelocityV1 | 0.18±0.02 | 0.21±0.14 | 0.38±0.15 | 0.03±0.03 | 0.13±0.04 | 0.14±0.19 |
| HalfCheetahVelocityV1 | 0.50±0.04 | 0.00±0.00 | 0.49±0.03 | 0.00±0.00 | 0.39±0.07 | 0.00±0.00 |
| Walker2dVelocityV1 | 0.13±0.03 | 1.50±0.40 | 0.11±0.04 | 1.26±0.59 | 0.05±0.03 | 0.25±0.09 |
| AntVelocityV1 | 0.59±0.00 | 0.00±0.00 | 0.53±0.01 | 0.00±0.00 | 0.29±0.05 | 0.00±0.00 |
| Average | 0.16 | 0.77 | 0.16 | 0.70 | 0.07 | 0.87 |

| Task | $H = 5$ | | Deepseek R1 | | Gemini 2.5 Pro | |
|---|---|---|---|---|---|---|
| | r↑ | c↓ | r↑ | c↓ | r↑ | c↓ |
| PointButton1 | 0.21±0.13 | 10.36±1.13 | 0.05±0.01 | 2.31±0.57 | 0.01±0.02 | 1.18±0.64 |
| PointButton2 | 0.27±0.07 | 11.76±2.04 | 0.08±0.03 | 2.57±0.60 | 0.07±0.02 | 1.80±0.27 |
| PointGoal1 | 0.06±0.13 | 5.19±3.41 | 0.35±0.03 | 0.71±0.24 | 0.31±0.02 | 0.63±0.20 |
| PointGoal2 | 0.26±0.06 | 6.49±3.44 | 0.05±0.03 | 0.11±0.08 | 0.10±0.04 | 0.68±0.31 |
| PointPush1 | 0.11±0.02 | 1.04±0.99 | 0.25±0.07 | 1.79±1.42 | 0.17±0.02 | 1.04±0.39 |
| PointPush2 | 0.10±0.02 | 1.69±2.18 | 0.12±0.02 | 0.82±0.48 | 0.10±0.01 | 0.51±0.35 |
| CarButton1 | -0.01±0.02 | 6.75±8.33 | -0.03±0.04 | 1.04±0.24 | -0.03±0.01 | 0.60±0.24 |
| CarButton2 | 0.03±0.01 | 13.28±4.62 | 0.00±0.01 | 1.94±0.34 | -0.03±0.02 | 1.02±0.21 |
| CarGoal1 | 0.16±0.09 | 3.11±2.03 | 0.15±0.04 | 0.11±0.06 | 0.11±0.00 | 0.03±0.04 |
| CarGoal2 | 0.06±0.04 | 8.77±7.05 | 0.02±0.02 | 0.60±0.33 | 0.03±0.01 | 0.38±0.42 |
| CarPush1 | 0.18±0.00 | 1.57±1.23 | 0.22±0.07 | 0.85±0.47 | 0.13±0.02 | 0.49±0.08 |
| CarPush2 | 0.06±0.01 | 4.16±1.62 | 0.07±0.05 | 0.42±0.06 | 0.04±0.02 | 0.15±0.21 |
| SwimmerVelocityV1 | 0.04±0.02 | 0.00±0.00 | -0.01±0.01 | 0.00±0.00 | 0.00±0.03 | 0.00±0.00 |
| HopperVelocityV1 | 0.06±0.05 | 0.47±0.35 | 0.29±0.10 | 1.27±1.39 | 0.20±0.03 | 0.25±0.27 |
| HalfCheetahVelocityV1 | 0.42±0.07 | 0.00±0.00 | 0.08±0.02 | 0.00±0.00 | 0.54±0.01 | 0.00±0.00 |
| Walker2dVelocityV1 | 0.12±0.04 | 0.43±0.26 | 0.17±0.02 | 2.03±0.46 | 0.15±0.02 | 1.66±0.61 |
| AntVelocityV1 | 0.49±0.01 | 0.00±0.00 | 0.54±0.11 | 0.00±0.00 | 0.39±0.10 | 0.00±0.00 |
| Average | 0.15 | 4.41 | 0.14 | 0.97 | 0.14 | 0.61 |

# G  LIMITATIONS

Although this work effectively addresses the challenge of learning offline safe policies in scenarios with scarce or no unsafe data, some limitations remain. The first lies in the accuracy and generalization of the environment model. When model accuracy is low, the rollout horizon must be restricted. A potential solution is to employ more powerful generative models, such as diffusion models, as environment models. The second limitation concerns the generation of conservative cost functions: the cost description must be grounded in features observable in the agent's observations. If certain variables required to compute the cost function cannot be directly or indirectly inferred from the observations, our approach cannot be applied. Meanwhile, despite the adoption of a check-and-feedback mechanism, the reliability of LLM outputs remains affected by hallucinations. A promising direction for future research is to integrate human-in-the-loop strategies to mitigate this issue. Finally, due to the reliance on cost function generation, the current PROCO is restricted to state-based tasks and cannot be applied to vision-based tasks. This is because VLMs cannot directly produce sufficiently accurate image-based cost functions, and performing VLM-based evaluation for every sample would be prohibitively expensive.

