# OpenReview forum: "Proactive Cost Generation for Offline Safe Reinforcement Learning Without Unsafe Data"
_ICLR.cc/2026/Conference — ICLR 2026 Conference Withdrawn Submission_

### Official Review · Reviewer_oxxF · 2025-10-16

**Soundness:** 2
**Presentation:** 3
**Contribution:** 2
**Rating:** 2
**Confidence:** 4

**Summary:**

This paper studies the problem of learning a safe policy from an offline dataset that has few or no unsafe samples. The proposed algorithm, called PROCO, uses a learned dynamics model and a cost function generated by LLM to perform rollouts, synthesizing more unsafe samples. The algorithm is tested on Safety-Gymnasium benchmark against several offline safe RL baselines, and results show that PROCO is effective in improving safety.

**Strengths:**

1. The proposed algorithm is effective in improving safety.
2. The writing of the paper is clear.

**Weaknesses:**

1. Generating the cost function requires natural language specification, including task information and cost description. Such specifications relies on detailed knowledge of the environment and task-by-task design.
2. The dynamics model used for generating unsafe samples is learned on a dataset with few unsafe samples in the first place. This means that the generated unsafe samples are likely unreliable because they come from extrapolating the dataset.

**Questions:**

1. Is a natural language specification of a task always available? Does the construction of such specifications generalizable to different tasks?
2. From Appendix D.2, it seems that writing the task descriptions requires even more knowledge and labor than constructing the cost functions. Why is the LLM necessary? Why not just handcraft the cost function?
3. How accurate is the dynamics model on unsafe samples that are not in the dataset? Why can we believe these samples since they come from extrapolation, which we aim to avoid in offline RL?
4. From the experimental results, the rewards of PROCO are much lower than the baselines. Is this because PROCO is very conservative or this is already a pretty high reward for a safe policy?
5. How to choose the proportion of safe samples to be labeled as unsafe? How does this proportion affect reward and cost performance?

---

### Official Review · Reviewer_1Chq · 2025-10-26

**Soundness:** 2
**Presentation:** 2
**Contribution:** 2
**Rating:** 4
**Confidence:** 4

**Summary:**

This paper proposes PROCO, a proactive offline safe RL framework for datasets with few or no unsafe samples. It combines a learned dynamics model with LLM-based cost estimation to generate counterfactual unsafe samples for feasibility-guided policy learning. Experiments show that PROCO improves safety and reduces violations compared to existing offline safe RL methods.

**Strengths:**

The motivation is clear, and the authors provide some theoretical justification for the proposed conservative feasible Bellman operator.

**Weaknesses:**

1. The assumption that the learned dynamics model has bounded error is strict and unlikely to hold when the policy encounters out-of-distribution states, particularly in settings with only safe or limited unsafe data. In addition, the paper does not specify the assumptions on the learned reward function.

2. The usage of LLM seems unnecessary, given that a clear description of the task’s safety constraint is available (lines 130-133) and the cost-related features can already be derived from the agent’s observations (Appendix G).

3. The policy learning component appears incremental compared to FISOR, as the main difference lies only in the addition of an extra term in Eq. (12) when computing $\mathcal{L}_{Q_h}$. This limits the novelty and technical contribution of the proposed method.

**Questions:**

1. The authors mention several times that the dataset consists of no more than 100 unsafe transitions (Section 3, 5.1, and Appendix D.2). Why was this specific number chosen?

2. Regarding Assumption 4.7 on the learned dynamics, since the dataset mostly contains safe data with very limited unsafe samples, how can the model accurately capture or generate unsafe transitions that lie outside the dataset’s support?

3. The paper introduces LLM-based cost generation, but Appendix D.2 indicates that human experts provide feedback to the LLM outputs. Given that task safety constraints are already described and experts are involved, why not manually design the cost function instead of relying on the LLM?

4. The results in Table 1 indicate that the proposed method achieves very low normalized rewards (e.g., 0.14) while maintaining safety. How do the authors ensure that this conservativeness does not simply result from trivial behavior, such as the agent remaining stationary to avoid violations?

5. What is the ratio of generated unsafe data to the original safe data, and how does varying the number of generated unsafe samples influence the learned policy’s performance and safety–reward trade-off?

---

### Official Review · Reviewer_w6WT · 2025-10-31

**Soundness:** 3
**Presentation:** 3
**Contribution:** 3
**Rating:** 6
**Confidence:** 3

**Summary:**

This paper addresses offline safe reinforcement learning when training data contains few or no unsafe samples. The authors propose **PROCO**, a framework that: (1) learns a dynamics model from offline data, (2) leverages large language models (LLMs) to generate conservative cost functions from natural language safety constraint descriptions, and (3) performs model-based rollouts to synthesize counterfactual unsafe samples. The key insight is identifying "safe-but-infeasible" states—states currently satisfying constraints but inevitably leading to violations within a few steps. PROCO demonstrates 5× improvement in safety over baselines across 17 Safety-Gymnasium tasks.

**Strengths:**

- Addresses a practical and underexplored scenario where collecting unsafe data is expensive or prohibited in safety-critical applications
-  Innovative application of LLMs to generate conservative cost functions from natural language, with a validation-and-feedback mechanism to ensure reliability
- Provides formal analysis (Theorems 4.6, 4.9) showing how conservative cost functions reduce rollout horizon and model error impact on feasibility identification

**Weaknesses:**

- Despite check-and-feedback, still susceptible to LLM hallucinations; some tasks (Walker2dVelocity) failed to achieve 100% accuracy on unsafe samples due to observation-reality misalignment

- Model learning, LLM queries, and iterative validation add overhead not thoroughly discussed

- Requires careful tuning of conservativeness range [p_min, p_max] and multiple LLM queries (up to 10) may still fail to find satisfactory cost functions

**Questions:**

1. How does PROCO perform when the cost function structure is highly complex or requires multi-step reasoning? Can the LLM-generated cost functions generalize to scenarios not covered in the prompt examples?

2. How does the proportion of safe-but-infeasible states in the dataset affect performance? Is there an analysis showing when PROCO provides the most benefit over baselines?

3. Figure 4(b) shows sharp performance degradation at H=5. Can you provide model prediction error analysis to explain why H must be restricted to 1? Does this limit the types of safety constraints PROCO can handle?

---

### Official Review · Reviewer_8q3P · 2025-11-01

**Soundness:** 2
**Presentation:** 2
**Contribution:** 2
**Rating:** 2
**Confidence:** 3

**Summary:**

The paper introduces PROCO, an offline safe Reinforcement Learning (RL) framework designed for scenarios where the pre-collected dataset contains few or no unsafe samples. Traditional offline safe RL methods struggle without risky data, often misclassifying “safe-but-infeasible” states as safe. PROCO addresses this by combining a learned dynamics model with LLM-generated conservative cost functions based on natural-language safety descriptions. It proactively simulates potential unsafe scenarios to identify infeasible states and guide safe policy learning. Experiments on 17 Safety-Gymnasium tasks show that PROCO significantly reduces safety violations compared to baselines.

**Strengths:**

- It addresses the novel and challenging problem of learning safe policies from offline datasets that contain few or no unsafe samples, which is a real-world, high-stakes scenario.

- The empirical results show improved safety performance.

- It introduces an interesting application of LLMs to generate a conservative cost function from a natural language description, adapting a known technique (LLMs for reward generation) to the safety domain.

**Weaknesses:**

- The paper's assumption that feasibility information is missing from the dataset is questionable. The premise states data is truncated by "external interventions" to prevent violations, which implies the final state of such trajectories is already known to be infeasible. This provides the exact feasibility signal that the paper's LLM and rollout components are designed to synthetically create.

- PROCO's core policy learning algorithm is structurally almost identical to FISOR. The paper's primary contribution is limited to a data pre-processing step: it first strips the original data of all cost signals and then adds back synthetic cost signals via LLM-based cost generation and model-based rollouts, which the PROCO, basically FISOR, algorithm can then use.

- The fundamental theoretical basis (multi-step reachability analysis for feasibility) is largely abandoned in practice as the default implementation uses an extremely limited rollout length ($\mathbf{H=1}$).

- The paper introduces a novel hard-constraint, "safe-only" problem, but tests it on the DSRL benchmark, which is designed for soft, cumulative constraints where violations are recoverable and expected in the data. This creates a fundamental mismatch. The paper's setup artificially removes cost signals to fit this premise, which is problematic for the chosen baselines. Soft-constraint methods like CPQ are unsuited for this zero-signal, hard-constraint task. Even FISOR, which does use hard constraints, is unfairly benchmarked, as its original design relies on observing cost signals to learn feasibility signals that were intentionally removed in PROCO's baseline comparison.

- Despite pursuing a hard constraint objective targeting zero cost, the resulting policy does not achieve this goal, exhibiting mostly non-zero violation rates.

- Coptidice and Fisor not sota OSRL baselines. COptiDICE, in particular, demonstrates weak safety performance in the DSRL paper. The literature review also appears incomplete, omitting several relevant OSRL publications, including:
  - OASIS [https://arxiv.org/pdf/2407.14653]
  - Constraint-Adaptive Policy Switching [https://www.arxiv.org/pdf/2412.18946]
  - Trajectory Classification for Safe RL [https://arxiv.org/pdf/2412.15429]
  - Constraint-conditioned actor-critic for OSRL [https://openreview.net/pdf?id=nrRkAAAufl]

- Tables 5 and 6 are hard to parse.

**Questions:**

- Please check weaknesses.
- How exactly were the safe-only datasets constructed? were entire trajectories removed or only violating transitions?
- In Figure 2, are the component ablations averaged over all tasks or shown for a single task, and if the latter, which task?

---

### Note · Authors · 2025-11-12

I have read and agree with the venue's withdrawal policy on behalf of myself and my co-authors.